# Beyond Confidence: Exploiting Homogeneous Pattern for Semi-Supervised Semantic Segmentation

Rui Sun [1] [*]   Huayu Mai [2] [3] [*]   Wangkai Li [2] [3]   Yujia Chen [2] [3]   Naisong Luo [2]   Yuan Wang [2]   Tianzhu Zhang [2] [3]

## Abstract

The critical challenge of semi-supervised semantic segmentation lies in how to fully exploit a large volume of unlabeled data to improve the model's generalization performance for robust segmentation. Existing methods mainly rely on confidence-based scoring functions in the prediction space to filter pseudo labels, which suffer from the inherent trade-off between true and false positive rates. In this paper, we carefully design an agent construction strategy to build clean sets of correct (positive) and incorrect (negative) pseudo labels, and propose the Agent Score function (AgScore) to measure the consensus between candidate pixels and these sets. In this way, AgScore takes a step further to capture homogeneous patterns in the embedding space, conditioned on clean positive/negative agents stemming from the prediction space, without sacrificing the merits of the confidence score, yielding a better trade-off. We provide a theoretical analysis to understand the mechanism of AgScore, and demonstrate its effectiveness by integrating it into three semi-supervised segmentation frameworks on Pascal VOC, Cityscapes, and COCO datasets, showing consistent improvements across all data partitions.

## 1. Introduction

Semantic segmentation, which aims to predict a specific semantic class for each pixel, has achieved conspicuous achievements attributed to the recent advances in deep neural network (Long et al., 2015) in computer vision with widespread applications such as embodied intelligence (Everingham et al., 2010; Wu et al., 2022), autonomous driving (Asgari Taghanaki et al., 2021), *etc*. However, its data-driven nature makes it labor-intensive and time-consuming to gather massive pixel-level annotations as training data. For example, it takes around 1.5 hours to label a single image on Cityscapes (Cordts et al., 2016) with merely 19 classes. To alleviate the data-hunger issue, considerable works (Yang et al., 2022a; Sun et al., 2023c; Hu et al., 2021) have turned their attention to semi-supervised semantic segmentation task. Since only limited labeled data is accessible, how to fully exploit a large amount of unlabeled data to improve the model's generalization performance for robust segmentation is thus extremely challenging.

To leverage unlabeled data effectively, consistency regularization (Sajjadi et al., 2016; Laine & Aila, 2016) and pseudo-labeling (Lee et al., 2013; Rizve et al., 2021) have emerged as mainstream paradigms for semi-supervised segmentation. Recently, these two paradigms are often assembled in the form of a teacher-student scheme (Wang et al., 2022a; Chen et al., 2023). In this scheme, the teacher network, with a weakly augmented view, generates *pseudo labels* to instruct the learning of the student model under a strongly augmented view. The success of this scheme relies on the correctness of the pseudo labels, as the quality of the selected pseudo labels determines the upper bound of performance. Therefore, it is crucial to design an appropriate criterion (referred to as *scoring function*) to detect and reject incorrect pseudo labels with low scores.

Recently, confidence-based methods (Sohn et al., 2020; Mai et al., 2024b; Na et al., 2023; Yang et al., 2022b), such as UniMatch (Yang et al., 2022a), have dominated this field credited to their simplicity and competitive performance. The mainstream practice is to attempt to set a hard threshold (*e.g.*, 0.95) as a scoring function to filter out pixel-level pseudo labels with low confidence in the prediction space. However, this scoring function tends not to be preferred ascribed to the inherent trade-off between the true positive rate (TPR) and false positive rate (FPR), as illustrated in Figure 1 (a). On the one hand, a high confidence threshold ensures the quality of pseudo labels (low FPR), but it discards numerous unconfident yet correct pseudo labels (unfavorably low

[*]Equal contribution [1]Shenzhen International Graduate School, Tsinghua University [2]MoE Key Laboratory of Brain-inspired Intelligent Perception and Cognition, University of Science and Technology of China [3]National Key Laboratory of Deep Space Exploration, Deep Space Exploration Laboratory. Correspondence to: Tianzhu Zhang <tzzhang@ustc.edu.cn>.

*Proceedings of the 42nd International Conference on Machine Learning*, Vancouver, Canada. PMLR 267, 2025. Copyright 2025 by the author(s).

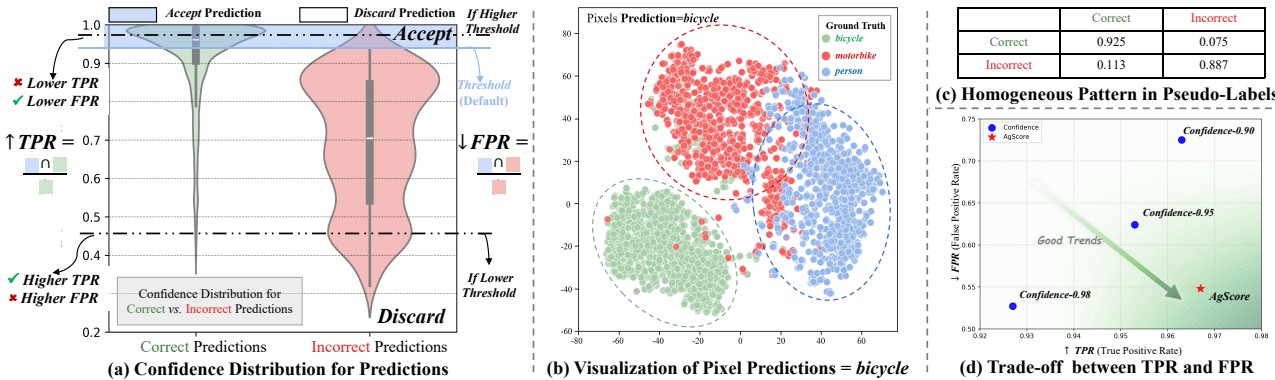

*Figure 1.* Motivation of the proposed AgScore. (a) Confidence distributions for correct and incorrect predictions, showing the inherent trade-off between TPR and FPR when using confidence thresholds. (b) Visualization of pixel embeddings predicted as the class *bicycle*, demonstrating the homogeneous pattern where correct and incorrect pseudo-labels form distinct clusters. (c) Quantitative analysis of the homogeneous pattern, indicating that pixels with higher similarity to correct pseudo-labels are more likely to be correctly predicted. (d) AgScore leverages the homogeneous pattern to achieve a better balance. The experiments are conducted on the 1/16 partition of the Pascal VOC dataset (Everingham et al., 2010), assuming that the ground truth for unlabeled data is available solely for theoretical analysis.

TPR), compromising the model's capability. On the other hand, lowering the threshold encourages the utilization of more correct pseudo-label pixels (high TPR) but inevitably enrolls erroneous pseudo labels that may mislead training (undesirably high FPR). Concurrently, several methods attempt to develop learning strategies to mine unlabeled data, including contrastive learning (Xu et al., 2022), disentangled supervision signals (Jin et al., 2022), and relation-wise consistency regularization (Mai et al., 2024b), etc. These methods operate in the embedding/feature space, which has been widely proven to have better tolerance for incorrect pseudo-labels compared to the prediction space. However, they are still confined to selecting pseudo labels in the prediction space (confidence scoring function), failing to tap into the potential of scoring pseudo labels from the embedding space itself. Then, the question naturally arises: *How to explore the better scoring function from the embedding space beyond confidence?*

Our motivation is derived from an intuitive fact: pixels belonging to similar patterns tend to share homogeneous semantics compared to different patterns. For example, Figure 1 (b) showcases the embedding distribution of correct and incorrect pseudo labels predicted as the class *bicycle*. It is evident that pixels from correct pseudo labels exhibit a propensity to congregate together, forming homogeneous patterns (*i.e.*, green dotted circle). This phenomenon can also be observed, but with more nuanced paradigms, in pixels from incorrect pseudo-labels, Incorrectly predicted classes often exhibit diverse patterns; for example, the classes *motorbike* and *person*, when erroneously classified as *bicycle*, correspond to two distinct patterns (red/blue dotted circle). We term this observation as *homogeneous pattern*. This provides a special bonus; in essence,

different patterns encode the relative semantic comparability, which can be reflected by the differences in pattern consensus, and is from a different perspective than the confidence scoring function in prediction space. We further quantitatively investigate the static consensus of correct and incorrect pseudo-label embeddings across different classes in Figure 1 (c). The results indicate that for any given pixel, there exists a higher probability of being correctly predicted if it exhibits a higher similarity to the set of correct pseudo labels compared to the set of incorrect pseudo labels. This provides hints for assessing the reliability of pseudo labels in the embedding space.

In this paper, we carefully design a new scoring function to filter out unreliable pseudo labels from a fresh perspective of the homogeneous pattern in embedding space beyond confidence, which is in line with the nature of the dense prediction task, for robust semi-supervised semantic segmentation. The main idea is to construct clean sets of correct and incorrect pseudo labels and leverage the homogeneous pattern derived from the noise-resistant embedding space to examine the consensus (similarity) differences between the candidate pixels and these sets in a nuanced and detailed manner. In specific, we devise an agent construction strategy with two considerations: (1) For constructing a clean set of correct pseudo labels (referred to as *positive agents*), it is easy to select high-quality agents equipped with a high confidence, considering that they have relatively simple patterns. (2) For constructing a high quality set of incorrect pseudo labels (*negative agents*), intuitively, these agents should resonate favorably with diverse semantic cues, considering that incorrect pseudo labels originate from multiple classes corresponding to diverse patterns. We prepend an orthogonal selection strategy that maintains sufficient dif-

ferences from positive agents without compromising quality. This strategy selects the most representative negative agents to cover a broader semantic space and enhances the evaluation capacity. Now, we are prepared to propose the Agent Score function (**AgScore**) to score pseudo labels, which is positively correlated with the similarity between candidate pseudo labels and positive agents and negatively correlated with the similarity to negative agents. In this manner, AgScore takes a step further to capture homogeneous patterns in the embedding space, conditioned on clean positive/negative agents stemming from the prediction space, without sacrificing the merits of confidence, yielding higher TPR and lower FPR (Figure 1 (d)). Furthermore, we provide a theoretical analysis to aid in understanding the mechanism of AgScore. This theoretical analysis translates into strong empirical performance when integrating AgScore into three semi-supervised segmentation methods including FixMatch (Sohn et al., 2020), UniMatch (Yang et al., 2022a) and RankMatch (Mai et al., 2024b), achieving consistent improvements for each across all data partitions in the Pascal VOC (Everingham et al., 2010), Cityscapes (Cordts et al., 2016), and COCO (Chollet, 2017) datasets.

In this work, our contributions can be summarized as follows: (1) We propose a novel Agent Score function (AgScore) that assesses the reliability of pseudo labels by exploiting the homogeneous pattern phenomenon, which is in line with the nature of the dense prediction task. AgScore measures the similarity between candidate pseudo labels and carefully constructed positive and negative agent sets to achieve a better trade-off. (2) We design an agent construction strategy including a confidence-based selection for positive agents and an orthogonal selection for negative agents. This strategy ensures the quality and diversity of the agent sets, enhancing AgScore's evaluation capability. (3) We provide a theoretical analysis to understand the mechanism of AgScore. Extensive experimental results on challenging benchmarks show that integrating AgScore into existing semi-supervised segmentation frameworks yields consistent improvements.

## 2. Related Work

**Semi-supervised Learning.** With the development of deep learning (Sun et al., 2023a;d;b; Mai et al., 2024a; Chen et al., 2025), semi-supervised learning (Zhu, 2005; Grandvalet & Bengio, 2004; Oliver et al., 2018) (SSL) has attracted increasing attention for reducing annotation costs. Recent research can be summarized in two branches: pseudo-labeling and consistency regularization. Pseudo-labeling (Lee et al., 2013; Cascante-Bonilla et al., 2021; Arazo et al., 2020; Zhang et al., 2021) methods involve training the model on unlabeled samples using pseudo-labels generated from the most up-to-date optimized model. On the other hand, consis-

tency regularization-based (Laine & Aila, 2016; Tarvainen & Valpola, 2017; Verma et al., 2022; Xie et al., 2020) methods leverage the smoothness assumption (Luo et al., 2018), encouraging the model to exhibit consistency when presented with the same example under different perturbations. Notably, recent SSL methods (Sohn et al., 2020; Berthelot et al., 2019; 2021; Xu et al., 2021; Guo & Li, 2022) have demonstrated the synergy between consistency regularization and Pseudo-labeling. Interestingly, recent SSL methods have demonstrated the synergy between consistency regularization and pseudo-labeling, combining the strengths of both approaches to achieve state-of-the-art performance. One prominent example is FixMatch (Sohn et al., 2020), which generates pseudo-labels from weakly augmented unlabeled images and uses them to train the model on strongly augmented versions of the same images. This approach effectively combines the benefits of pseudo-labeling, which provides additional training targets, with the regularization effects of consistency training, which encourages the model to be robust to perturbations. This concise yet powerful approach has gained widespread adoption in recent SSL studies.

**Semi-supervised Semantic Segmentation.** Benefits from the advances in deep neural network (Sun et al., 2021; Wang et al., 2022b; Mai et al., 2023; Luo et al., 2023; Wang et al., 2023d) and various kinds of semi-supervised semantic segmentation (SSSS) algorithms (Zhao et al., 2023b;a; Liang et al., 2023; Sun et al., 2023c; Na et al., 2023; Liu et al., 2023; Howlader et al., 2024; 2025; Mai et al., 2025; Sun et al., 2025) have been proposed based on the mature combination of Pseudo-labeling and consistency regularization. One notable work in this direction is UniMatch (Yang et al., 2022a), which takes into account the nature of semantic segmentation tasks and incorporates suitable data augmentations into the popular FixMatch framework. By carefully designing augmentations that preserve the semantic content of the images while introducing meaningful variations, UniMatch has evolved into a concise yet powerful SSSS baseline. The success of UniMatch has inspired numerous follow-up studies, which have further refined and extended this approach to achieve state-of-the-art performance on various benchmark datasets. On top of these fundamental designs, motivated by representation learning, a series of works (Wang et al., 2023c;a; 2022c; Ma et al., 2023) have incorporated contrastive learning into SSSS, tailoring it to the characteristics of the dense prediction task. For example, Alonso et al. (Alonso et al., 2021) introduce a memory bank to store high-quality class features and perform positive-only contrastive learning. By maintaining a set of representative features for each class and using them as positive samples during contrastive learning, this approach can effectively capture the essential characteristics of different semantic categories and improve the model's discriminative power.

Similarly, PC$^2$Seg (Zhong et al., 2021) adopts pixel-level contrastive learning and introduces several negative sampling techniques to avoid the problem of sampling error. By carefully selecting negative samples that are likely to be informative and diverse, PC$^2$Seg can learn more robust and generalizable representations for semantic segmentation. Despite the significant progress made in SSSS research, there remain several challenges and open questions. One key challenge is the presence of noisy or incorrect pseudo-labels, which can mislead the model and hinder its performance. Previous methods tend to use pseudo-label filtering strategies based on confidence scores. Instead of being trapped in focusing on confidence-based scoring functions, we underline that homogeneous pattern is also worth investigating beyond confidence.

## 3. Method

In this section, we first formulate the semi-supervised semantic segmentation problem as preliminaries and introduce the core idea of our AgScore from the perspective of homogeneous pattern. Then we describe the details of the agent construction strategy and propose a natural scoring function to detect the reliability of the candidate pixel predictions. Finally, the theoretical analysis is provided from the perspective of multi-label classification to demonstrate the effectiveness of our AgScore.

### 3.1. Preliminaries

Given a labeled set $\mathcal{D}^l = \{(\boldsymbol{x}_i^l, \boldsymbol{y}_i^l)\}_{i=1}^{N^l}$ and an unlabeled set $\mathcal{D}^u = \{\boldsymbol{x}_i^u\}_{i=1}^{N^u}$, where $N^u \gg N^l$, semi-supervised semantic segmentation aims to train a segmentation model with limited labeled data and vast unlabeled data. The popular teacher-student scheme consists of a teacher network $f_T$ and a student network $f_S$. The student network is guided by two sources of supervision, including the ground truth for the labeled data and the pseudo labels generated by the teacher network for the unlabeled data. The teacher network can either be identical to the student network or an exponentially moving average (EMA) version of the student network. Specifically, for the labeled data, the supervised loss $\mathcal{L}_{sup}$ can be formulated as:

$$\mathcal{L}_{sup} = \frac{1}{N^l} \sum_{i=1}^{N^l} \frac{1}{HW} \sum_{j=1}^{HW} \ell_{ce}\left(\boldsymbol{y}_{ij}^l, f_S(\boldsymbol{x}_i^l)_j\right), \quad (1)$$

where $H$ and $W$ represent the height and width of the input image, $\ell_{ce}$ denotes the standard pixel-wise cross-entropy loss. For the unlabeled data, the dominant confidence-based methods set a high threshold (*e.g.*, 0.95) as a scoring function to filter out pseudo labels (generated by the teacher

network) with low confidence:

$$\hat{\boldsymbol{y}}_{ij}^u = \begin{cases} \arg\max f_T(aug(\boldsymbol{x}_i^u))_j, & c_{ij}^u > \gamma \\ \text{ignore\_index}, & \text{otherwise} \end{cases}, \quad (2)$$

where $c_{ij}^u = \max f_T(aug(\boldsymbol{x}_i^u))_j$ represents the confidence of the teacher prediction for $j^{th}$ pixel and $\gamma$ denotes the confidence threshold to exclude unreliable pseudo labels from training. As result, we can obtain the consistency regularization loss $\mathcal{L}_{reg}$ as:

$$\mathcal{L}_{reg} = \frac{1}{N^u} \sum_{i=1}^{N^u} \frac{1}{HW} \sum_{j=1}^{HW} \ell_{ce}\left(\hat{\boldsymbol{y}}_{ij}^u, f_S(\mathcal{A}ug(\boldsymbol{x}_i^u))_j\right),$$
$$(3)$$

where $\mathcal{A}ug(\cdot)$ means the strong augmentation. By imposing consistency regularization, the model can learn reliable information from unlabeled data. The overall loss of the commonly used teacher-student scheme is $\mathcal{L} = \mathcal{L}_{sup} + \mathcal{L}_{reg}$.

While these methods have shown promising results, the scoring function they employ is often not favored due to its disregard for many correct pseudo labels with low confidence in the early stages of training, raised by the strict threshold. As training progresses, they indiscriminately increase confidence in all pseudo labels, eventually recruiting over-confident erroneous pseudo labels into training and undermining the model's performance. In this paper, we carefully design a new scoring function to filter out unreliable pseudo labels and retain reliable pseudo labels from a fresh perspective of homogeneous pattern. What follows, we detail the agent construction strategy first.

### 3.2. Agent Construction

To enable the model to detect the reliability of candidate pixel predictions, we construct clean sets of correct and incorrect pseudo labels and leverage extra clues to examine the consensus (similarity) differences between the candidate pixels and these sets. We operate at the level of pixel embeddings to enable the calculation of similarity between candidate pixels and the agent sets. Specifically, we obtain the pixel embeddings $\boldsymbol{F} \in \mathbb{R}^{B \times HW \times C}$ for a batch of unlabeled image $\{\boldsymbol{x}_i^u\}_{i=1}^B$ before the classifier of the segmentation model and construct positive and negative agent sets from it. The overall agent construction process is shown in Algorithm 1.

For the clean set of pixels with correct pseudo labels (referred to as *positive agents* $\{\boldsymbol{a}_n^p\}_{n=1}^N$), considering that they have relatively simple patterns, we randomly select $N$ pixels from the top-1% confidence pixels in $\boldsymbol{F}$ in a class-balanced manner and obtain $\{\boldsymbol{a}_n^p\}_{n=1}^N$.

For the clean set of pixels with incorrect pseudo labels (referred to as *negative agents* $\{\boldsymbol{a}_m^n\}_{m=1}^M$), considering that incorrect pseudo labels originate from multiple classes corresponding to diverse patterns, we design an orthogonal

*Table 1.* Quantitative results of different SSL methods on PASCAL *classic* set. We report mIoU (%) under various partition protocols and show the improvements $\Delta$ over the baseline.

| Method | ResNet-50 | | | | | ResNet-101 | | | | |
|---|---|---|---|---|---|---|---|---|---|---|
| | 1/16 (92) | 1/8 (183) | 1/4 (366) | 1/2 (732) | Full (1464) | 1/16 (92) | 1/8 (183) | 1/4 (366) | 1/2 (732) | Full (1464) |
| *Sup.-only* | 44.0 | 52.3 | 61.7 | 66.7 | 72.9 | 45.1 | 55.3 | 64.8 | 69.7 | 73.5 |
| PCR[NeurIPS'22] (Xu et al., 2022) | – | – | – | – | – | 70.0 | 74.7 | 77.1 | 78.5 | 80.7 |
| GTA-Seg[NeurIPS'22] (Jin et al., 2022) | – | – | – | – | – | 70.0 | 73.2 | 75.6 | 78.4 | 80.5 |
| ReCo[ICLR'22] (Liu et al., 2021) | 64.8 | 72.0 | 73.1 | 74.7 | – | – | – | – | – | – |
| AugSeg[CVPR'23] (Zhao et al., 2023b) | 64.2 | 72.1 | 76.1 | 77.4 | 78.8 | 71.0 | 75.4 | 78.8 | 80.3 | 81.3 |
| NP-SemiSeg[ICML'23] (Wang et al., 2023b) | 65.7 | 72.3 | 75.7 | 77.4 | – | – | – | – | – | – |
| DAW[NeurIPS'23] (Sun et al., 2023c) | 68.5 | 73.1 | 76.3 | 78.6 | 79.7 | 74.8 | 77.4 | 79.5 | 80.6 | 81.5 |
| DDFP[CVPR'24] (Wang et al., 2024) | – | – | – | – | – | 74.9 | 78.0 | 79.5 | 81.2 | 81.9 |
| PRCL[IJCV'24] (Xie et al., 2024) | – | – | – | – | – | 71.2 | 72.2 | 75.2 | 76.2 | 78.3 |
| FixMatch[NeurIPS'21] (Sohn et al., 2020) | 60.1 | 67.3 | 71.4 | 73.7 | 76.9 | 63.9 | 73.0 | 75.5 | 77.8 | 79.2 |
| **FixMatch+AgScore** | 63.2 | 69.6 | 73.1 | 75.3 | 77.9 | 67.0 | 75.5 | 77.5 | 79.5 | 80.4 |
| $\Delta\uparrow$ | +3.1 | +2.3 | +1.7 | +1.6 | +1.0 | +3.1 | +2.5 | +2.0 | +1.7 | +1.2 |
| UniMatch[CVPR'23] (Yang et al., 2022a) | 67.4 | 71.9 | 75.3 | 78.0 | 79.3 | 73.5 | 75.4 | 78.7 | 80.2 | 81.9 |
| **UniMatch+AgScore** | 69.4 | 73.6 | 76.8 | 79.4 | 80.0 | 75.6 | 77.2 | 80.1 | 81.3 | 82.6 |
| $\Delta\uparrow$ | +2.0 | +1.7 | +1.5 | +1.4 | +0.7 | +2.1 | +1.8 | +1.4 | +1.1 | +0.7 |
| RankMatch[CVPR'24] (Mai et al., 2024b) | 71.6 | 74.6 | 76.7 | 78.8 | 80.0 | 75.5 | 77.6 | 79.8 | 80.7 | 82.2 |
| **RankMatch+AgScore** | 73.1 | 75.8 | 77.7 | 79.9 | 80.4 | 76.1 | 78.2 | 80.5 | 81.2 | 82.6 |
| $\Delta\uparrow$ | +1.5 | +1.2 | +1.0 | +1.1 | +0.4 | +0.6 | +0.6 | +0.5 | +0.5 | +0.4 |

selection strategy. Specifically, as shown in Algorithm 2, we first select those pixels with very low confidence (bottom-1%) and then incrementally build the negative agents set $\{a_m^n\}_{m=1}^M$ sampled from the low-confidence pixels set such that a new agent is maximally orthogonal (*i.e.*, minimal cosine similarity) to the agents already selected, starting with a pixel features at random, where $M$ denotes the number of negative agents. This strategy selects the most representative negative agents to cover more semantic space and enhances the evaluation ability of reliability.

### 3.3. Agent Score

Now, we are prepared to propose the Agent Score function (AgScore) to assess the reliability of pseudo labels. Intuitively, the AgScore is supposed to be positively correlated with the similarity between candidate pseudo labels and positive agents and negatively correlated with the similarity to negative agents. A natural design is to use a sum-softmax function to check the proportion of the similarity between the pixel $\boldsymbol{f}$ and positive agents $\{a_n^p\}_{n=1}^N$ in the union of positive agents and negative agents $\{a_n^p\}_{n=1}^N \cup \{a_m^n\}_{m=1}^M$:

$$S(\boldsymbol{f}) = \frac{\sum_{n=1}^N e^{cos(\boldsymbol{f}, \boldsymbol{a}_n^p)}}{\sum_{n=1}^N e^{cos(\boldsymbol{f}, \boldsymbol{a}_n^p)} + \sum_{m=1}^M e^{cos(\boldsymbol{f}, \boldsymbol{a}_m^n)}}. \quad (4)$$

By modeling homogeneous pattern beyond confidence-based scores, noisy pseudo labels will be suppressed while the reliable ones will be highlighted, thus increasing their involvement in consistency regularization-based training. Finally, inspired by DivideMix (Li et al., 2020), we fit a two-component GMM for $S$ using the Expectation-Maximization algorithm. The Gaussian component with a larger mean corresponds to the correct ones. For each pixel $i$, the probability of being correct is given by the posterior probability $P(\text{correct} \mid S_i)$. When the $P(\text{correct} \mid S_i) > 0.5$, the pseudo-label of that pixel will be used in the consistency regularization, instead of following the confidence thresholding strategy in Equation 2.

## 4. Theoretical Analysis

To better understand how AgScore can facilitate pseudo-label selection, we provide a theoretical analysis from the perspective of multi-label classification to demonstrate that the separability of correct and incorrect pseudo labels can be improved with the help of positive and negative agents.

*Table 2.* Quantitative results of different SSL methods on PASCAL *blender* set. We report mIoU (%) under various partition protocols and show the improvements Δ over the baseline.

| Method | ResNet-50 | | | ResNet-101 | | |
|---|---|---|---|---|---|---|
| | 1/16 (662) | 1/8 (1323) | 1/4 (2646) | 1/16 (662) | 1/8 (1323) | 1/4 (2646) |
| *Sup.-only* | 62.4 | 68.2 | 72.3 | 67.5 | 71.1 | 74.2 |
| AEL[NeurIPS'21] (Hu et al., 2021) | – | – | – | 77.2 | 77.6 | 78.1 |
| PCR[NeurIPS'22] (Xu et al., 2022) | – | – | – | 78.6 | 80.7 | 80.8 |
| GTA-Seg[NeurIPS'22](Jin et al., 2022) | – | – | – | 77.8 | 80.5 | 80.6 |
| AugSeg[CVPR'23] (Zhao et al., 2023b) | 74.7 | 76.0 | 77.2 | 77.0 | 77.3 | 78.8 |
| CFCG[ICCV'23] (Li et al., 2023a) | 75.0 | 77.1 | 77.7 | 76.8 | 79.1 | 79.9 |
| NP-SemiSeg[ICML'23] (Wang et al., 2023b) | 73.4 | 76.5 | 76.7 | – | – | – |
| DAW[NeurIPS'23] (Sun et al., 2023c) | 76.2 | 77.6 | 77.4 | 78.5 | 78.9 | 79.6 |
| DDFP[CVPR'24] (Wang et al., 2024) | – | – | – | 78.3 | 78.8 | 79.8 |
| PRCL[IJCV'24] (Xie et al., 2024) | – | – | – | 77.9 | 79.1 | 79.9 |
| FixMatch[NeurIPS'21] (Sohn et al., 2020) | 70.6 | 73.9 | 75.1 | 74.3 | 76.3 | 76.9 |
| **FixMatch+AgScore** | 73.7 | 76.9 | 77.9 | 77.7 | 79.2 | 78.7 |
| Δ ↑ | +3.1 | +3.0 | +2.8 | +3.4 | +2.9 | +1.8 |
| UniMatch[CVPR'23] (Yang et al., 2022a) | 75.8 | 76.9 | 76.8 | 78.1 | 78.4 | 79.2 |
| **UniMatch+AgScore** | 77.8 | 78.8 | 77.9 | 80.1 | 80.3 | 80.5 |
| Δ ↑ | +2.0 | +1.9 | +1.1 | +2.0 | +1.9 | +1.3 |
| RankMatch[CVPR'24] (Mai et al., 2024b) | 76.6 | 77.8 | 78.3 | 78.9 | 79.2 | 80.0 |
| **RankMatch+AgScore** | 78.4 | 79.1 | 79.2 | 80.0 | 80.3 | 80.6 |
| Δ ↑ | +1.8 | +1.3 | +0.9 | +1.1 | +1.1 | +0.6 |

## 4.1. Discrete Version of AgScore

For theoretical tractability, we consider a discrete version of AgScore, denoted as:

$$\hat{S} = \frac{X}{X + Y}, \tag{5}$$

where $X$ and $Y$ represent the count of positive and negative agents exceeding a certain similarity threshold $\psi$, respectively. Specifically, we define:

$$X = \sum_{n=1}^{N} \mathbb{1}[s_n \geq \psi], \quad s_n = e^{\cos(\boldsymbol{f}, \boldsymbol{a}_n^p)}, \tag{6}$$

$$Y = \sum_{m=1}^{M} \mathbb{1}[s_m \geq \psi], \quad s_m = e^{\cos(\boldsymbol{f}, \boldsymbol{a}_m^n)}. \tag{7}$$

In this context, $X$ represents the number of positive agents whose similarity to pixel $\boldsymbol{f}$ exceeds the threshold, corresponding to a multi-label classification task on the positive agents. Similarly, $Y$ represents the count of the negative agents. When $X$ is relatively large, the pixel $\boldsymbol{f}$ exhibits higher similarity to the positive agents, making it more likely to be a correct pseudo-label, which corresponds to a larger $\hat{S}$, and vice versa. This characteristic indicates that the discrete version of AgScore behaves consistently with its

soft counterpart, making our subsequent analysis practically meaningful.

Note that $\hat{S} = 1/(1 + Z)$, where $Z = Y/X$, is a *monotonic* function of $Z$. Studying $Z$ simplifies derivations while still reflecting the underlying distributional differences between correct and incorrect pseudo labels. Therefore, we focus on the ratio $Z = Y/X$.

## 4.2. Distribution Approximation

We assume that the similarity scores $s_n = e^{\cos(\boldsymbol{f}, \boldsymbol{a}_n^p)}$ follow an implicit distribution, leading to a Bernoulli random variable:

$$\hat{s}_n = \mathbb{1}[s_n \geq \psi] \sim \text{Bernoulli}(p), \tag{8}$$

where $p = P(s_n \geq \psi)$. Similarly, $\hat{s}_m = \mathbb{1}[s_m \geq \psi] \sim$ Bernoulli$(q)$, where $q = P(s_m \geq \psi)$[1]. According to binomial approximation rules, we have the following lemma:

**Lemma 4.1.** *For a given pixel $\boldsymbol{f}$, the counts of highly simi-*

---

[1]Actually, since $p/q$ varies with the agents, $X$ and $Y$ follow a Poisson binomial distribution. We will prove in the Appendix C that the subsequent derivations hold universally.

*Table 3.* Quantitative results of different SSL methods on Cityscapes. We report mIoU (%) under various partition protocols and show the improvements $\Delta$ over the baseline.

| Method | ResNet-50 | | | | ResNet-101 | | | |
|---|---|---|---|---|---|---|---|---|
| | 1/16 (186) | 1/8 (372) | 1/4 (744) | 1/2 (1488) | 1/16 (186) | 1/8 (372) | 1/4 (744) | 1/2 (1488) |
| *Sup.-only* | 63.3 | 70.2 | 73.1 | 76.6 | 66.3 | 72.8 | 75.0 | 78.0 |
| AEL[NeurIPS'21] (Hu et al., 2021) | 74.0 | 75.8 | 76.2 | – | 75.8 | 77.9 | 79.0 | 80.3 |
| PCR[NeurIPS'22] (Xu et al., 2022) | – | – | – | – | 73.4 | 76.3 | 78.4 | 79.1 |
| GTA-Seg[NeurIPS'22] (Jin et al., 2022) | – | – | – | – | 69.4 | 72.0 | 76.1 | – |
| AugSeg[CVPR'23] (Zhao et al., 2023b) | 73.7 | 76.5 | 78.8 | 79.3 | 75.2 | 77.8 | 79.5 | 80.4 |
| Co-Train[ICCV'23] (Li et al., 2023b) | – | 76.3 | 77.1 | – | 75.0 | 77.3 | 78.7 | – |
| NP-SemiSeg[ICML'23] (Wang et al., 2023b) | 73.0 | 77.1 | 78.8 | 78.7 | – | – | – | – |
| DAW[NeurIPS'23] (Sun et al., 2023c) | 75.2 | 77.5 | 79.1 | 79.5 | 76.6 | 78.4 | 79.8 | 80.6 |
| DDFP[CVPR'24] (Wang et al., 2024) | – | – | – | – | 77.1 | 78.1 | 79.8 | 80.8 |
| PRCL[IJCV'24](Xie et al., 2024) | – | – | – | – | 73.4 | 77.0 | 77.9 | 80.0 |
| FixMatch[NeurIPS'21] (Sohn et al., 2020) | 72.6 | 75.7 | 76.8 | 78.2 | 74.2 | 76.2 | 77.2 | 78.4 |
| **FixMatch+AgScore** | 76.5 | 78.3 | 78.9 | 80.4 | 77.6 | 79.0 | 79.2 | 79.5 |
| $\Delta\uparrow$ | +3.9 | +2.6 | +2.1 | +2.2 | +3.4 | +2.8 | +2.8 | +1.1 |
| UniMatch[CVPR'23] (Yang et al., 2022a) | 75.0 | 76.8 | 77.5 | 78.6 | 76.6 | 77.9 | 79.2 | 79.5 |
| **UniMatch+AgScore** | 76.8 | 78.4 | 78.8 | 79.2 | 78.3 | 78.4 | 79.9 | 80.2 |
| $\Delta\uparrow$ | +1.8 | +1.6 | +1.3 | +0.6 | +1.7 | +5.0 | +7.0 | 79.5 |
| RankMatch[CVPR'24] (Mai et al., 2024b) | 75.4 | 77.7 | 79.2 | 79.5 | 77.1 | 78.6 | 80.0 | 80.7 |
| **RankMatch+AgScore** | 76.6 | 78.6 | 79.7 | 79.9 | 78.2 | 79.6 | 80.5 | 81.0 |
| $\Delta\uparrow$ | +1.2 | +0.9 | +0.5 | +0.4 | +1.1 | +1.0 | +0.5 | +0.3 |

*lar positive and negative agents satisfy:*

$$X = \sum_{n=1}^{N} \hat{s}_n \sim Binomial(N, p) \xrightarrow{d} \mathcal{N}(Np, Np(1-p)),$$
(9)

$$Y = \sum_{m=1}^{M} \hat{s}_m \sim Binomial(M, q) \xrightarrow{d} \mathcal{N}(Mq, Mq(1-q)),$$
(10)

*where $\xrightarrow{d}$ denotes converges in distribution.*

Based on this lemma, we can derive the distribution of $Z$:

**Proposition 4.2.** *The ratio $Z = Y/X$ follows a log-normal distribution:*
$$Z \xrightarrow{d} \mathcal{LN}(\mu, \sigma^2),$$
(11)

*where*

$$\mu = \ln \frac{Mq}{Np}, \quad \sigma^2 = \frac{1-q}{Mq} + \frac{1-p}{Np}.$$
(12)

*Proof.* See Appendix D. □

Let $p_1$ and $p_2$ represent the probabilities of pixels with correct and incorrect pseudo labels that share high similarity with *positive* agents, respectively. It follows that $p_1 > p_2$ naturally. Similarly, $q_1 < q_2$ holds. Consequently, for correct pseudo labels, $Z_1 \sim \mathcal{LN}(\mu_1, \sigma_1^2)$, and for incorrect pseudo labels, $Z_2 \sim \mathcal{LN}(\mu_2, \sigma_2^2)$. With the distributions of correct and incorrect pseudo labels established, we can now conduct a quantitative analysis of their separability.

### 4.3. Separability of Pseudo-label

To quantify the separability between correct and incorrect pseudo labels, we analyze the false positive rate (FPR) at a fixed true positive rate (TPR), *i.e.*, $\lambda$.

**Lemma 4.3.** *In the form of log-normal distribution, the $\mathrm{FPR}_\lambda$ can be represented as:*

$$\mathrm{FPR}_\lambda = F_2(F_1^{-1}(\lambda))$$
$$= \frac{1}{2}\left\{1 + \mathrm{erf}\left[\frac{\sigma_1}{\sigma_2}\mathrm{erf}^{-1}(2\lambda - 1) + \frac{\mu_1 - \mu_2}{\sqrt{2}\sigma_2}\right]\right\}$$
(13)

*where $F_1$ and $F_2$ denote the cumulative distribution functions of $Z_1$ and $Z_2$, respectively.*

*Proof.* See Appendix E □

Note that the number of positive agents $N$ is closely related to TPR: increasing $N$ would increase the probability of the existence of highly similar positive agents, thus increasing TPR. However, our metric $\text{FPR}_\lambda$ is defined at a fixed TPR. So we keep $N$ constant and focus on the effect of $M$.

**Proposition 4.4.** *For a fixed $N$, $\text{FPR}_\lambda$ is a decreasing function of $M$, i.e.,*

$$\frac{\partial \text{FPR}_\lambda}{\partial M} = \frac{e^{-t^2}}{\sqrt{\pi}} \cdot \left[ \text{erf}^{-1}(2\lambda - 1)\frac{\partial \frac{\sigma_1}{\sigma_2}}{\partial M} + \frac{1}{\sqrt{2}}\frac{\partial \frac{\mu_1 - \mu_2}{\sigma_2}}{\partial M} \right]$$
$$< 0. \tag{14}$$

*Proof.* See Appendix F. $\qquad \square$

This suggests that increasing $M$ improves the separability between correct and incorrect pseudo labels, which aligns with our observation that the patterns of incorrect pseudo labels are quite diverse. Therefore, in our experiments, we opted for a relatively large $M$ (with $M > N$) to ensure the effectiveness of AgScore. However, this does not imply that a higher number of negative agents is always better. As the number of negative agents increases, the newly added negative agents will inevitably have semantic overlap with the positive sample agents, leading to reduced differences between positive and negative agents, which hinders the model's learning.

# 5. Experiments

## 5.1. Experimental Setup

**Datasets:** (1) **PASCAL VOC 2012** (Everingham et al., 2010) is an object-centric semantic segmentation dataset, containing 20 object classes in the foreground and a background class with 1,464 and 1,449 finely annotated images for training and validation, respectively. Many researches (Chen et al., 2021; Hu et al., 2021) augment the original training set (*i.e.*, *classic*) with additional 9,118 coarsely annotated images in SBD (Hariharan et al., 2011) to get a *blender* training set. (2) **Cityscapes** (Cordts et al., 2016) is an urban scene understanding dataset consisting of 2,975 images for training and 500 images for validation. The initial 30 semantic classes are re-mapped into 19 classes for the semantic segmentation task. (3) **COCO** (Lin et al., 2014), composed of 118k/5k training/validation images, containing 81 classes to predict.

**Implementation Details:** For a fair comparison, we use ResNet-50/101 (He et al., 2016) pretrained on ImageNet (Krizhevsky et al., 2012) as the backbone and DeepLabv3+ (Chen et al., 2018) as the decoder. The crop size is set as $513 \times 513$ for PASCAL and $801 \times 801$ for

Cityscapes, respectively. We adopt stochastic gradient descent (SGD) optimizer with an initial learning rate of 0.001 for PASCAL and 0.005 for Cityscapes. Polynomial Decay learning rate policy is applied throughout the whole training. Further, when training a baseline integrated with our method, we use the same weak and strong augmentations as used by the corresponding baseline. The features used to construct the correlation consistency are extracted from the output of the ASPP module (Chen et al., 2017) and the channel number is 256. We set the number of positive agents $N = 64$ and the number of negative agents $M = 256$ for all experiments. The model is trained for 80 epochs on PASCAL and 240 epochs on Cityscapes with a batch size of 8, using $8\times$ RTX 3090 GPUs (memory is 24G/GPU).

## 5.2. Comparison with State-of-the-art Methods

We integrate AgScore into three representative SSL frameworks: FixMatch (Sohn et al., 2020), UniMatch (Yang et al., 2022a), RankMatch (Mai et al., 2024b), and evaluate its performance with both ResNet-50 and ResNet-101 backbones under various partition protocols, following common practices in the field.

**Results on PASCAL.** Table 1 presents the comparative results of AgScore with state-of-the-art approaches on the PASCAL VOC *classic* set. Our method consistently enhances the performance of all baseline SSL frameworks across all partition protocols and backbone networks. Notably, under the most challenging $1/16$ partition with only 92 labeled images, AgScore significantly boosts the performance of FixMatch, UniMatch, and RankMatch by $3.1\%$, $2.1\%$, and $0.6\%$, respectively, with ResNet-101 backbone, demonstrating its powerful capability in leveraging unlabeled data under extremely label-scarce scenarios. Similar improvements are observed on the PASCAL VOC *blender* set, as shown in Table 2. AgScore consistently improves upon the baseline methods across all settings, with the most significant gains achieved under the $1/16$ partition, further validating its effectiveness.

**Results on Cityscapes.** Table 3 shows the comparative results on the Cityscapes dataset. Our AgScore continues to enhance the performance of all baseline methods across various partition protocols and backbone networks. Under the most challenging $1/16$ partition with only 186 labeled images, AgScore boosts the performance of FixMatch, UniMatch, and RankMatch by $3.4\%$, $1.7\%$, and $1.1\%$ respectively with ResNet-101 backbone, verifying its robustness and generalization ability.

**Results on COCO.** Table 4 summarizes that AgScore consistently achieves improvement across all partitions, demonstrating the effectiveness of integrating the proposed AgScore into the UniMatch framework. The improvement is particularly significant in the low-data regime, validating

AgScore's ability, especially when labeled data is scarce. The results align with the motivation and theoretical analysis, showcasing the potential of exploring scoring functions beyond confidence in the embedding space to address the inherent trade-off between true positive rate and false positive rate in pseudo-label selection.

**Qualitative Results.** We compare the qualitative results of our method with different SOTA methods on the PASCAL dataset in Figure 2. AgScore shows more powerful segmentation performance in fine-grained details (e.g., the dogs on the bed and the man on horseback). With the help of modeling homogeneous patterns beyond confidence, AgScore exhibits superior abilities in most scenarios.

### 5.3. Ablation Study and Analysis

To look deeper into our method, we perform a series of ablation studies on PASCAL *classic* set under $1/16(92)$ partition protocol with ResNet-50 to analyze our AgScore.

**Analysis of AgScore.** In Table 5, we report the results of $1/16(92)$ and Full(1464) to clearly substantiate the effectiveness of our design. Note that, "*baseline*" denotes the reproduced results for UniMatch (Yang et al., 2022a). We have the following findings: (1) With the increase in the number of negative agents M, our AgScore achieves consistent performance improvements, which can be attributed to our design of orthogonal selection strategy that maintains sufficient differences from positive agents without compromising quality. This strategy selects the most representative negative agents to cover more semantic space and enhances the ability to evaluate reliability (index 2-5, index 6-7). (2) However, this does not imply that a higher number of negative agents is always better. As the number of negative agents increases, the newly added negative agents will inevitably have semantic overlap with the positive sample agents, leading to reduced differences between positive and negative agents, which hinders the model's learning. This aligns with our theoretical explanation (index 4-5, index 7-8). (3) Within a certain range, increasing the number of positive samples does not lead to significant improvements in performance (index 2-5 *vs.*index 6-8). This is because it is easy to select high-quality agents equipped with high confidence, considering that they have relatively simple patterns. However, when the number of samples exceeds a specific limit, it can harm performance due to the introduction of incorrect but high-confidence agents, which affects the purity of the positive set (index 9-10).

**Analysis of Agent Construction Strategy.** In Table 6, we explore various strategies for agent construction, including "Uniform" representing uniformly sampling negative agents from the candidate set, "Bottom" representing sampling negative agents in ascending order of confidence. (1) Uniformly selecting candidate agents is not a desirable approach, as it inevitably introduces considerable noise among these agents. This noise can adversely affect the quality of pseudo labels, resulting in sub-optimal performance. (2) The strategy of "Orthogonal" achieves the best results with a light computational cost, which is in line with our design purpose, with light computational cost.

**Trade-off of Pseudo Labels.** As shown in Figure 3, in practice, AgScore determines the criteria for selecting pseudo-labels by exploiting the homogeneous pattern phenomenon, which is in line with the nature of the dense prediction task and is conducive to model training. In this way, the model improves, benefiting from the effective probing of reliable pseudo-labels. And in turn, the positive distribution will be maximally separated from the negative agents. In a nutshell, AgScore takes a step further to capture homogeneous patterns in the embedding space, conditioned on clean positive/negative agents stemming from the prediction space, without sacrificing the merits of confidence, yielding higher TPR and lower FPR.

## 6. Conclusion

In this paper, we proposed AgScore, a novel scoring function that exploits the homogeneous pattern phenomenon to assess the reliability of pseudo labels for semi-supervised semantic segmentation. AgScore measures the similarity between candidate pseudo labels and carefully constructed positive and negative agent sets, highlighting reliable pseudo labels while suppressing noisy ones. Extensive experimental results on challenging benchmarks show the effectiveness.

### Acknowledgments

This work was partially supported by the National Defense Science and Technology Foundation Strengthening Program Funding (Grant 2023-JCJQ-JJ-0219).

### Impact Statement

This paper presents work whose goal is to advance the field of Machine Learning. There are many potential societal consequences of our work, none of which we feel must be specifically highlighted here.

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

# A. Algorithm

---

**Algorithm 1** Agent Construction.

---

**Inputs:** Pixel Embeddings $\boldsymbol{F} \in \mathbb{R}^{B \times HW \times C}$, Prediction $\boldsymbol{P} \in \mathbb{R}^{B \times HW}$, #Positive Agents $N$, #Negative Agents $M$
**Output:** Positive Agents $\boldsymbol{A}^p \in \mathbb{R}^{N \times C}$, Negative Agents $\boldsymbol{A}^n \in \mathbb{R}^{M \times C}$

 1: $K = \text{Unique\_Classes}(\boldsymbol{P})$             $\triangleright$ Determine the predicted classes set
 2: Initialize empty sets $\boldsymbol{F}^p, \boldsymbol{F}^n$
 3: **for** each class $k \in K$ **do**
 4:    $\boldsymbol{F}_k = \{\boldsymbol{f}_{ij} \in \boldsymbol{F} \mid \boldsymbol{P}(\boldsymbol{f}_{ij}) = k\}$        $\triangleright$ Extract pixels predicted as class $k$
 5:    $\boldsymbol{F}_k^p = \{\boldsymbol{f}_{ij} \in \boldsymbol{F}_k \mid c_{ij}^u \in \text{top-}\frac{1}{|K|}\%\}$
 6:    $\boldsymbol{F}_k^n = \{\boldsymbol{f}_{ij} \in \boldsymbol{F}_k \mid c_{ij}^u \in \text{bottom-}\frac{1}{|K|}\%\}$
 7:    $\boldsymbol{F}^p = \boldsymbol{F}^p \cup \boldsymbol{F}_k^p$
 8:    $\boldsymbol{F}^n = \boldsymbol{F}^n \cup \boldsymbol{F}_k^n$
 9: **end for**
10: $\boldsymbol{A}^p = \text{Random\_Select}(\boldsymbol{F}^p, N)$
11: $\boldsymbol{A}^n = \text{Orthogonal\_Selection}(\boldsymbol{F}^n, M)$
12: **return** $\boldsymbol{A}^p, \boldsymbol{A}^n$

---

**Algorithm 2** Orthogonal Selection Strategy.

---

**Inputs:** Pixel Embedding $\boldsymbol{F} \in \mathbb{R}^{M \times C}$, #Agents $N$
**Output:** Agents $\boldsymbol{A} \in \mathbb{R}^{N \times C}$

 1: $\boldsymbol{A} = \text{Random\_Select}(\boldsymbol{F}, 1)$              $\triangleright$ Randomly initialized
 2: **for** $i = 1$ **to** $N - 1$ **do**
 3:    $\boldsymbol{F} = \{\boldsymbol{f} \mid \boldsymbol{f} \in \boldsymbol{F}, \boldsymbol{f} \notin \boldsymbol{A}\}$           $\triangleright$ Difference Set
 4:    $\boldsymbol{S} = \text{Cosine\_Similarity}(\boldsymbol{F}, \boldsymbol{A})$       $\triangleright \boldsymbol{S} \in \mathbb{R}^{(HW-i) \times i}$
 5:    $\boldsymbol{S} = \text{Max}(\boldsymbol{S}, \text{dim=1})$
 6:    $\text{index} = \text{Argmin}(\boldsymbol{S}, \text{dim=0})$         $\triangleright$ Min-Max Strategy
 7:    $\boldsymbol{a} = \text{Select}(\boldsymbol{F}, \text{index})$
 8:    $\boldsymbol{A} = \boldsymbol{A} \cup \boldsymbol{a}$
 9: **end for**
10: **return** $\boldsymbol{A}$

---

## B. More Results

In this section, we provide more experimental results due to space constraints in the main paper. These include:

(1) Detailed results on the COCO datasets (Table 4), showing consistent improvements from AgScore across different data partitions and baseline models. (2) Additional qualitative visualizations on PASCAL VOC (Figure 2) that further demonstrate the advantages of AgScore, particularly in challenging regions. (3) Ablation studies on hyperparameter selection and the impact of different agent selection strategies (Tables 5 and 6), providing more insights to support the conclusions in the main text. (4) Analysis of trade-off between TPR and FPR comparing AgScore to confidence thresholding during training (Figure 3).

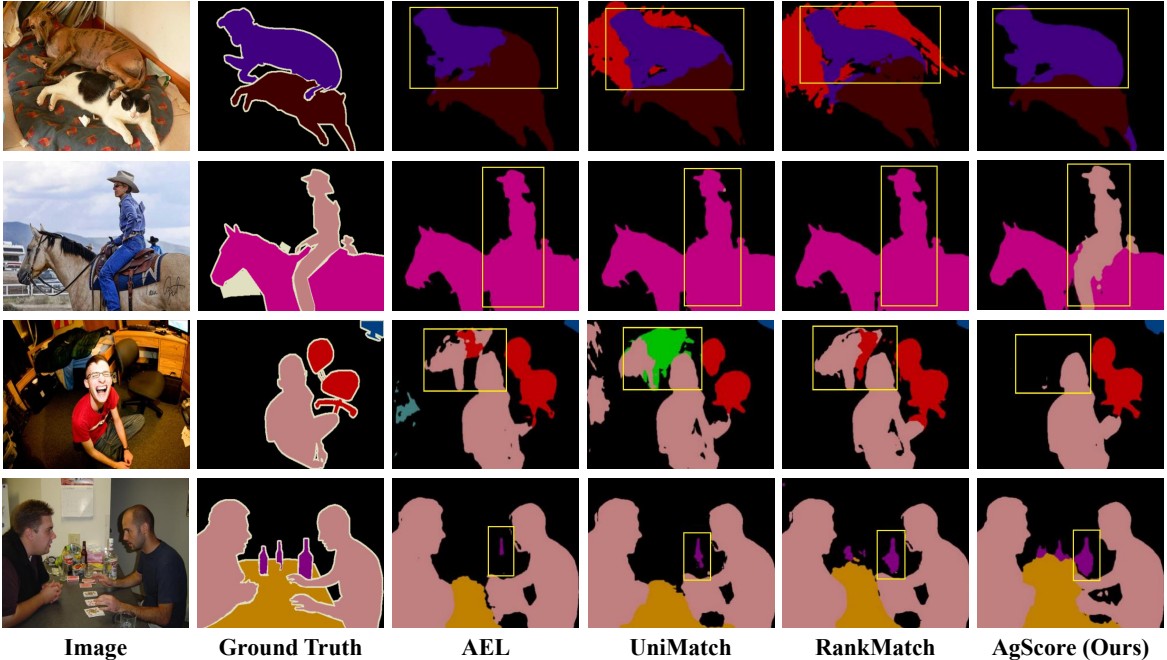

| Image | Ground Truth | AEL | UniMatch | RankMatch | AgScore (Ours) |

*Figure 2.* Qualitative comparison with different methods. Note that significant improvements are marked with yellow boxes.

*Table 4.* Quantitative results of different SSL methods on COCO. We report mIoU (%) under various partition protocols and show the improvements $\Delta$ over the baseline.

| Method | 1/512 (232) | 1/256 (463) | 1/128 (925) | 1/64 (1849) | 1/32 (3697) |
|---|---|---|---|---|---|
| *Sup.-only* | 22.9 | 28.0 | 33.6 | 37.8 | 42.2 |
| PseudoSeg[Arxiv'20] (Zou et al., 2020) | 29.8 | 37.1 | 39.1 | 41.8 | 43.6 |
| PC2Seg[ICCV'21] (Zhong et al., 2021) | 29.9 | 37.5 | 40.1 | 43.7 | 46.1 |
| CISC-R[TPAMI'23] (Wu et al., 2023) | 32.1 | 40.2 | 42.2 | – | – |
| UniMatch[CVPR'23] (Yang et al., 2022a) | 31.9 | 38.9 | 44.4 | 48.2 | 49.8 |
| **UniMatch+AgScore** | 33.9 | 40.6 | 45.7 | 49.7 | 51.2 |
| $\Delta \uparrow$ | +2.0 | +1.7 | +1.3 | +1.5 | +1.4 |

*Table 5.* Quantitative results of the effectiveness of AgScore.

|    | Variants |        | mIoU(92) | mIoU(1464) |
|----|----------|--------|----------|------------|
| 1  | Baseline |        | 67.4     | 79.3       |
| 2  |          | M=64   | 68.5     | 79.6       |
| 3  | N=64     | M=128  | 69.0     | 79.7       |
| 4  |          | M=256  | 69.4     | 80.0       |
| 5  |          | M=512  | 68.1     | 79.4       |
| 6  |          | M=128  | 69.2     | 79.6       |
| 7  | N=128    | M=256  | 69.5     | 80.2       |
| 8  |          | M=512  | 67.8     | 79.6       |
| 9  | N=256    | M=256  | 68.1     | 79.2       |
| 10 |          | M=512  | 67.7     | 79.0       |

*Table 6.* The effectiveness of agent construction strategy.

| Method            | mIoU(92) | Flops (G) |
|-------------------|----------|-----------|
| Baseline          | 67.4     | 17.4      |
| Uniform           | 67.7     | 17.5      |
| Bottom            | 68.0     | 17.5      |
| **Orthogonal (ours)** | **69.4** | **18.7** |

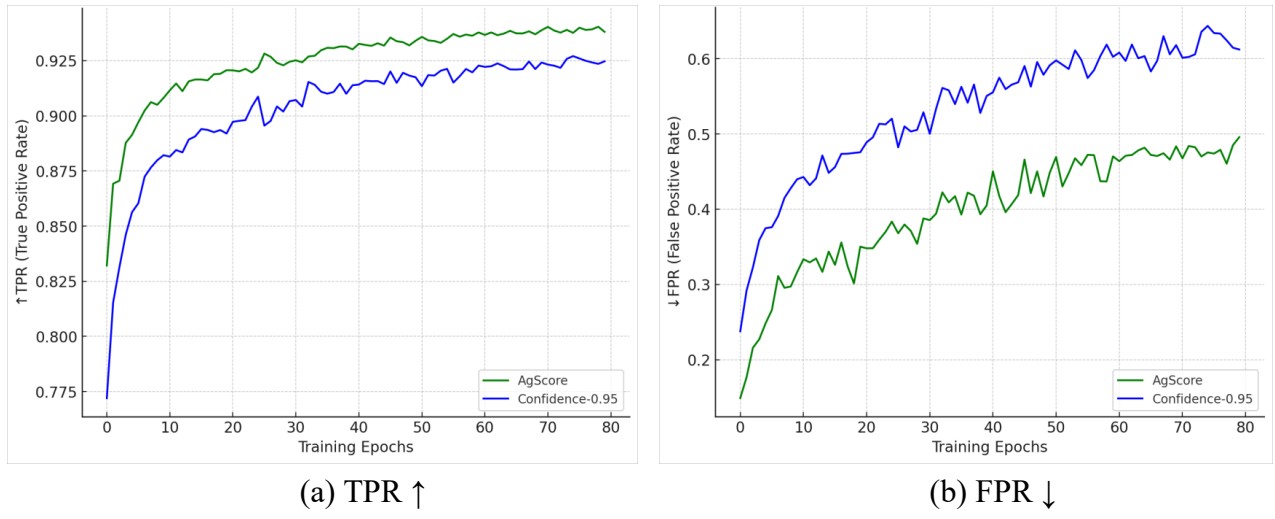

(a) TPR ↑                                          (b) FPR ↓

*Figure 3.* Comparison of AgScore and Confidence-0.95 in terms of (a) True Positive Rate (TPR) and (b) False Positive Rate (FPR) over training epochs.

## C. More General Case of $X$ & $Y$

In the previous derivation, we treated all $P(s_n \geq \psi)$ as equal to a constant $p$, allowing us to sum independent and identically distributed Bernoulli random variables to obtain a binomial distribution, which we then approximated as a Gaussian distribution for the sake of convenience in subsequent derivations. However, in reality, $p_n = P(s_n > \psi)$ should vary with the agents and the sum follows the Poisson binomial distribution. In this section, we will demonstrate that, under more general conditions, $X$ and $Y$ can still be approximated as Gaussian distributions. This implies that, despite the simplifications made for the convenience of derivation, the subsequent derivations remain valid and can effectively reflect the separability between correct and incorrect pseudo-labels.

Formally, rewrite $X = \sum_{n=1}^{N} \hat{s}_n$, where $\hat{s}_n = \mathbb{1}[s_n \geq \psi] \sim \text{Bernoulli}(p_n)$, $p_n = P(s_n \geq \psi)$. It is easily shown that:

$$\mathbb{E}[\hat{s}_n] = p_n, \quad \text{Var}[\hat{s}_n] = p_n(1 - p_n). \tag{15}$$

$$\mathbb{E}[X] = \sum_{n=1}^{N} p_n, \quad \text{Var}[X] = \sum_{n=1}^{N} p_n(1 - p_n). \tag{16}$$

Since $\hat{s}_n$ can only take values 0 and 1, we have:

$$
\begin{aligned}
\mathbb{E}[|\hat{s}_n - \mathbb{E}[\hat{s}_n]|^{2+\delta}] &= |1 - p_n|^{2+\delta} P(\hat{s}_n = 1) + |0 - p_n|^{2+\delta} P(\hat{s}_n = 0) \\
&= (1 - p_n)^{2+\delta} p_n + p_n^{2+\delta}(1 - p_n) \\
&< 2
\end{aligned}
\tag{17}
$$

$$
\begin{aligned}
(\text{Var}[X])^{\frac{2+\delta}{2}} &= \left(\sum_{n=1}^{N} p_n(1 - p_n)\right)^{\frac{2+\delta}{2}} \\
&\geq \left(\sum_{n=1}^{N} r\right)^{\frac{2+\delta}{2}} \\
&= r^{\frac{2+\delta}{2}} N^{\frac{2+\delta}{2}},
\end{aligned}
\tag{18}
$$

where $r = \min\{p_n(1 - p_n)\}$. Following Equation 17 and Equation 18, we have:

$$0 < \frac{1}{(\text{Var}[X])^{\frac{2+\delta}{2}}} \sum_{n=1}^{N} \mathbb{E}[|\hat{s}_n - \mathbb{E}[\hat{s}_n]|^{2+\delta}] < \frac{2N}{r^{\frac{2+\delta}{2}} N^{\frac{2+\delta}{2}}} = \frac{2}{r^{\frac{2+\delta}{2}} N^{\frac{\delta}{2}}} \tag{19}$$

Obviously, $X$ satisfies the Lyapunov condition:

$$\frac{1}{(\text{Var}[X])^{\frac{2+\delta}{2}}} \sum_{n=1}^{N} \mathbb{E}[|\hat{s}_n - \mathbb{E}[\hat{s}_n]|^{2+\delta}] \to 0 \quad \text{as } N \to \infty, \tag{20}$$

Based on the Lyapunov central limit theorem, we have:

$$X \xrightarrow{d} \mathcal{N}(\mathbb{E}[X], \text{Var}[X]). \tag{21}$$

Similarly, $Y$ also approximates a Gaussian distribution.

## D. Proof of Proposition 4.2

Rewrite the Proposition:

**Proposition D.1.** *The ratio $Z = Y/X$ follows a log-normal distribution:*

$$Z \xrightarrow{d} \mathcal{LN}(\mu, \sigma^2), \tag{22}$$

*where*

$$\mu = \ln \frac{Mq}{Np}, \quad \sigma^2 = \frac{1-q}{Mq} + \frac{1-p}{Np}. \tag{23}$$

*Proof.* Taking the logarithm on both sides, we have:

$$\ln Z = \ln Y - \ln X. \tag{24}$$

To apply the Delta method, we note that since $X$ and $Y$ are approximately normally distributed, we can compute the mean and variance of $\ln X$ and $\ln Y$.

For a positive random variable $X$ that is approximately normally distributed, we can use the Delta method to derive:

$$\mathbb{E}[\ln X] \approx \ln \mathbb{E}[X],$$

and

$$\mathrm{Var}[\ln X] \approx \frac{\mathrm{Var}[X]}{(\mathbb{E}[X])^2}.$$

Applying this to $X$ and $Y$, we have:

- For $X$:
$$\mathbb{E}[\ln X] \approx \ln(Np),$$
$$\mathrm{Var}[\ln X] \approx \frac{(1-p)}{Np}.$$

- For $Y$:
$$\mathbb{E}[\ln Y] \approx \ln(Mq),$$
$$\mathrm{Var}[\ln Y] \approx \frac{(1-q)}{Mq}.$$

Since the difference of two normally distributed variables is also normally distributed, it follows that $\ln Z$ is normally distributed. Thus, $Z$ is log-normally distributed:

$$Z \xrightarrow{d} \mathcal{LN}(\mu, \sigma^2), \tag{25}$$

where

$$\mu = \ln \frac{Mq}{Np}, \quad \sigma^2 = \frac{1-q}{Mq} + \frac{1-p}{Np}. \tag{26}$$

$\square$

# E. Proof of Lemma 4.3

Rewrite the Lemma:

**Lemma E.1.** *In the form of log-normal distribution, the* $\mathrm{FPR}_\lambda$ *can be represented as:*

$$
\begin{aligned}
\mathrm{FPR}_\lambda &= F_2(F_1^{-1}(\lambda)) \\
&= \frac{1}{2}\left\{1 + \mathrm{erf}\left[\frac{\sigma_1}{\sigma_2}\mathrm{erf}^{-1}(2\lambda - 1) + \frac{\mu_1 - \mu_2}{\sqrt{2}\sigma_2}\right]\right\}
\end{aligned}
\tag{27}
$$

*where $F_1$ and $F_2$ denote the cumulative distribution functions of $Z_1$ and $Z_2$, respectively.*

*Proof.* Based on Proposition 4.2,

- For correct pseudo-labels, $Z_1 \sim \mathcal{LN}(\mu_1, \sigma_1^2)$, where:

$$
\mu_1 = \ln\frac{Mq_1}{Np_1}, \quad \sigma_1^2 = \frac{1 - q_1}{Mq_1} + \frac{1 - p_1}{Np_1}.
\tag{28}
$$

- For incorrect pseudo-labels, $Z_2 \sim \mathcal{LN}(\mu_2, \sigma_2^2)$, where:

$$
\mu_2 = \ln\frac{Mq_2}{Np_2}, \quad \sigma_2^2 = \frac{1 - q_2}{Mq_2} + \frac{1 - p_2}{Np_2}.
\tag{29}
$$

The cumulative distribution function $F(x)$ and inverse function $F^{-1}(\lambda)$ can be represented as:

$$
F(x; \mu, \sigma^2) = \frac{1}{2}[1 + \mathrm{erf}(\frac{\ln x - \mu}{\sqrt{2}\sigma})]
\tag{30}
$$

$$
F^{-1}(\lambda; \mu, \sigma^2) = \exp\{\mu + \sqrt{2}\sigma\mathrm{erf}^{-1}(2\lambda - 1)\}
\tag{31}
$$

Since $p_1 > p_2$ and $q_1 < q_2$, we have $\mu_1 < \mu_2$. For a given TPR $\lambda$, the threshold $\tau$ can be derived:

$$
\tau = F^{-1}(\lambda; \mu_1, \sigma_1^2) = \exp\{\mu_1 + \sqrt{2}\sigma_1\mathrm{erf}^{-1}(2\lambda - 1)\}
\tag{32}
$$

For a given threshold $\tau$, the FPR can be obtained:

$$
\begin{aligned}
F(\tau; \mu_2, \sigma_2^2) &= \frac{1}{2}[1 + \mathrm{erf}(\frac{\ln \tau - \mu_2}{\sqrt{2}\sigma_2})] \\
&= \frac{1}{2}\left\{1 + \mathrm{erf}\left[\frac{\sigma_1}{\sigma_2}\mathrm{erf}^{-1}(2\lambda - 1) + \frac{\mu_1 - \mu_2}{\sqrt{2}\sigma_2}\right]\right\}
\end{aligned}
\tag{33}
$$

*i.e.,*

$$
\mathrm{FPR}_\lambda = \frac{1}{2}\left\{1 + \mathrm{erf}\left[\frac{\sigma_1}{\sigma_2}\mathrm{erf}^{-1}(2\lambda - 1) + \frac{\mu_1 - \mu_2}{\sqrt{2}\sigma_2}\right]\right\}
\tag{34}
$$

$\square$

# F. Proof of Proposition 4.4

Rewrite the Proposition 4.4:

**Proposition F.1.** *For a fixed $N$,* $\mathrm{FPR}_\lambda$ *is a decreasing function of* $M$, *i.e.,*

$$\frac{\partial \mathrm{FPR}_\lambda}{\partial M} = \frac{e^{-t^2}}{\sqrt{\pi}} \cdot \left[ \mathrm{erf}^{-1}(2\lambda - 1)\frac{\partial \frac{\sigma_1}{\sigma_2}}{\partial M} + \frac{1}{\sqrt{2}}\frac{\partial \frac{\mu_1 - \mu_2}{\sigma_2}}{\partial M} \right] \tag{35}$$
$$< 0.$$

*Proof.* We start from the formula:

$$\mathrm{FPR}_\lambda = \frac{1}{2}\left\{ 1 + \mathrm{erf}\left[ \frac{\sigma_1}{\sigma_2}\mathrm{erf}^{-1}(2\lambda - 1) + \frac{\mu_1 - \mu_2}{\sqrt{2}\sigma_2} \right] \right\}. \tag{36}$$

The error function $\mathrm{erf}(x)$ is defined as:

$$\mathrm{erf}(x) = \frac{2}{\sqrt{\pi}}\int_0^x e^{-t^2}dt. \tag{37}$$

Next, we differentiate $\mathrm{FPR}_\lambda$ with respect to $M$ using the chain rule. Let $t = \frac{\sigma_1}{\sigma_2}\mathrm{erf}^{-1}(2\lambda - 1) + \frac{\mu_1 - \mu_2}{\sqrt{2}\sigma_2}$:

$$\frac{\partial \mathrm{FPR}_\lambda}{\partial M} = \frac{e^{-t^2}}{\sqrt{\pi}} \cdot \frac{\partial}{\partial M}\left[ \frac{\sigma_1}{\sigma_2}\mathrm{erf}^{-1}(2\lambda - 1) + \frac{\mu_1 - \mu_2}{\sqrt{2}\sigma_2} \right]$$
$$= \frac{e^{-t^2}}{\sqrt{\pi}} \cdot \left[ \mathrm{erf}^{-1}(2\lambda - 1)\frac{\partial \frac{\sigma_1}{\sigma_2}}{\partial M} + \frac{1}{\sqrt{2}}\frac{\partial \frac{\mu_1 - \mu_2}{\sigma_2}}{\partial M} \right] \tag{38}$$

We obtain:

$$\frac{\partial \mathrm{FPR}_\lambda}{\partial M} = \frac{e^{-t^2}}{\sqrt{\pi}} \cdot \left[ \mathrm{erf}^{-1}(2\lambda - 1)\frac{\partial \frac{\sigma_1}{\sigma_2}}{\partial M} + \frac{1}{\sqrt{2}}\frac{\partial (\mu_1 - \mu_2)}{\partial M} \cdot \frac{1}{\sigma_2} \right]. \tag{39}$$

Next, we analyze the derivatives of these components.

1. **Regarding $\sigma_1$ and $\sigma_2$**: $\sigma_1$ and $\sigma_2$ represent the standard deviations of correct and incorrect pseudo-labels, respectively. As the number of negative agents $M$ increases, $\sigma_2$ increases due to the increased diversity of incorrect pseudo-labels, leading to a decrease in $\frac{\sigma_1}{\sigma_2}$. This implies $\frac{\partial \frac{\sigma_1}{\sigma_2}}{\partial M} < 0$.

2. **Regarding $\mu_1 - \mu_2$**: $\mu_1$ and $\mu_2$ are the means associated with correct and incorrect pseudo-labels, respectively. With an increase in $M$, $\mu_2$ may increase, while $\mu_1$ remains relatively stable, resulting in $\frac{\partial (\mu_1 - \mu_2)}{\partial M} < 0$.

Combining these parts, we conclude:

$$\frac{\partial \mathrm{FPR}_\lambda}{\partial M} < 0.$$

Thus, $\mathrm{FPR}_\lambda$ decreases as $M$ increases.

$\square$

