# OpenReview forum: "Beyond Confidence: Exploiting Homogeneous Pattern for Semi-Supervised Semantic Segmentation"
_ICML.cc/2025/Conference — ICML 2025 poster_

### Official Review · Reviewer_JJYk · 2025-03-11

**Overall Recommendation:** 3

**Summary:**

This article proposes a new metric, AgScore, for pseudo label filtering. It measures the accuracy of pseudo-labels by evaluating the similarity between a pixel's embedding and positive pixel embeddings, as well as the dissimilarity between it and negative pixel embeddings. This method can be integrated as a universal plugin into existing SSL frameworks and improves performance on various baseline models.

**Claims And Evidence:**

The empirical results and theory of this article reflect that AgScore is a better indicator for filtering pseudo labels.
But this paper fails to point out the reason why AgScore is superior to the vanilla confidence.

**Essential References Not Discussed:**

None.

**Experimental Designs Or Analyses:**

Yes I did.

**Methods And Evaluation Criteria:**

Yes I did.

**Other Comments Or Suggestions:**

None.

**Other Strengths And Weaknesses:**

Advantage:
1. promising performances
2. clear written

disadvantage:
1. Due to Universal Visual Large Model, this task is not as valuable as before.
2. The reason why AgScore can defeat vanilla confidence is not clear.

**Questions For Authors:**

Though I can find some technological differences, the design of the AgScore has similarity to some existing works (e.g., RankMatch[1]) that leverage inter-pixel relationships. Would you like me to elaborate on the essential distinctions between them?

For other questions, please see the Strengths and Weaknesses part.

[1] Zhang Z, Chen W, Fang C, et al. Rankmatch: Fostering confidence and consistency in learning with noisy labels[C]//Proceedings of the IEEE/CVF international conference on computer vision. 2023: 1644-1654.

**Relation To Broader Scientific Literature:**

The contribution of this paper lies in the more accurate indicator for filtering pseudo labels. However, there are two main issues.
1. Why is the AgScore better than vanilla confidence? The theoretical proof only indicates the rationality of AgScore, but cannot prove how it is better than existing metrics.
2. Due to the emergence of the Universal Visual Large Models (SAM series), by 2025, the SSL task will no longer be challenging and have practical value.

**Theoretical Claims:**

Yes I did.

---

> ### Author Rebuttal · Authors · 2025-04-01
>
> Thanks for taking the time to share your comments in the review assessment. We provide a detailed point-by-point response to your comments.
>
> Note that the following **link** refers to https://anonymous.4open.science/r/AgScore/Rebuttal.pdf
>
> ---
>
> **Q1**: Advantages of AgScore.
>
> **A1**: Semi-supervised semantic segmentation is essentially a learning strategy-centric task, and its core lies in effectively filtering pseudo-labels to explore unlabeled data.
>
> - Previous **confidence-based methods tends not to be preferred ascribed to the trade-off** between TPR and FPR when handling pseudo-labels. A high confidence threshold ensures pseudo-label quality (low FPR) but discards many correct pseudo-labels with lower confidence (unfavorably low TPR), and vice versa.
> - We focus on the homogeneous pattern in the embedding space. Intuitively, **the high-dimensional, sophisticated embedding space has greater potential for assessing pseudo-label reliability than the relatively simple one-dimensional confidence metric from the prediction space**. This provides hints for assessing pseudo-label reliability in the embedding space.
> - In implementation, we absorb the merits of confidence to construct clean positive and negative agents. For pixel predictions difficult for confidence to handle, we employ the agent score function to score pseudo-labels in collaboration with clean positive/negative agents, **leveraging the embedding space's capability without sacrificing confidence's advantages, yielding higher TPR and lower FPR**.
>
>
> Experimentally:
>
> - Fig. 1 (d) shows that AgScore achieves a better TPR-FPR balance over confidence.
> - Fig. 3 depicts the better TPR and FPR dynamics of AgScore than confidence-based methods during training.
>
> These experimental results, along with the superior segmentation performance (Tab. 1-4), provide strong evidence that AgScore reaches new heights in reliable pseudo label filtering beyond confidence.
>
>
> Furthermore, we attempt to analyze from the perspective of information theory. We prove that the mutual information between the pseudo labels selected by AgScore and the ground truth is greater than that of regular confidence-based methods (**Theorem G.3 in link**). Therefore, AgScore theoretically improves pseudo label selection compared to the confidence metric.
>
> In summary, from the perspectives of **intuitive motivation**, **experimental analysis**, and **theoretical justification**, we demonstrate the advantages of AgScore over confidence-based methods.
>
> ---
>
> **Q2**: About SAM.
>
> **A2**: We respectfully disagree with your viewpoint.
>
> - Semi-supervised learning focuses on better exploring vast amounts of unlabeled data under extremely limited labeled data.
> - In fact, even for powerful visual foundation models like SAM, they **struggle to generalize to domain-specific scenarios**, such as medical images, requiring fine-tuning to adapt to downstream tasks. In these tasks, obtaining sufficient annotations is challenging and time-consuming, making it highly valuable to explore how SAM can leverage easily accessible unlabeled data.
>
> To further validate the value of SSL, we evaluate AgScore with SAM under the extreme 1-labeled case. As shown in **link** Tab. 8, we find: (1) The original SAM model struggles under the extreme 1-shot setting, underscoring the challenge of domain transfer for large vision models. (2) Fine-tuning SAM with a confidence-based pseudo-labeling strategy significantly boosts performance, demonstrating the value of SSL for adapting foundation models to specialized tasks. (3) Integrating AgScore into the SAM fine-tuning pipeline further improves results, validating the effectiveness of AgScore even with powerful vision backbones.
>
> ---
>
> **Q3**: About RankMatch.
>
> **A3**: Thanks for pointing this out. RankMatch proposes a sample selection strategy via a confidence voting scheme in the embedding space to increase sample selection quantity for learning with noisy labels.
>
> However, we argue that a essential distinction is that **RankMatch focuses solely on positive samples** , neglecting the utilization of negative samples. Unlike RankMatch, **AgScore combines knowledge from both positive and negative samples** by examining the difference in embedding similarity between the candidate pseudo-label's corresponding pixel and the positive/negative agents to measure reliability, thus better leveraging the capabilities of the embedding space. Moreover, we provide a theoretical justification for AgScore's working mechanism.
>
> Furthermore, we construct an ablation study in **link** Tab. 9 to verify the impact of considering only positive agents (1-st entry), demonstrating that the introduction of negative agents improves performance.
>
> ---
>
> We hope our response can resolve your concern. Please do not hesitate to let us know if you have further questions :)
>
> Thanks for your time and consideration. Have a wonderful day!
>
> [1] How to Efficiently Adapt Large Segmentation Model (SAM) to Medical Images.

---

> > ### Comment · Reviewer_JJYk · 2025-04-07
> >
> > Thank the author for the thoughtful and detailed rebuttal. The reviewer sincerely appreciates the time and effort the authors put into addressing the concerns. After carefully reviewing your responses, The reviewer is pleased to see that all of doubts have been resolved.
> >
> > Thanks for the authors' thorough and detailed rebuttal. After reviewing the authors' response, I believe it has addressed some of my doubts. However, I still have a main concern:
> >
> > About the SAM part.
> > I agree with the author that VLMs like SAM struggle to generalize to domain-specific scenarios. Hence, I admit that semi-supervised learning, weakly-supervised learning, unsupervised learning, and domain-adaption are valuable in annotation-sparse scenes like medical images, remote sensing images, etc. But in the current scene understanding of natural images, these methods are no longer the most advanced mainstream solutions. Thus, the SSL segmentation in natural images is not a valuable task for me.
> >
> > Overall, I think this paper gives some methodological insight, thus I will improve my score.

---

> > > ### Author Response · Authors · 2025-04-09
> > >
> > > Dear Reviewer JJYk,
> > >
> > > Many thanks for your score improvement and your positive feedback! Your comments complete our paper and make it better!
> > >
> > > Regarding your question, we are happy to engage in further discussion.
> > >
> > > We agree that vision foundation models like SAM have advanced the segmentation field. However,  prompt-based SAM may not be well-suited to directly handle real-world scene understanding applications in natural domain (e.g., autonomous driving) without fine-tuning, due to the following reasons:
> > >
> > > - Distribution shift and input perturbations: Vision foundation models, including SAM, are known to *exhibit vulnerability* to real-world distribution shifts, such as compression-induced corruptions [1]. These may arise from varying JPEG compression rates depending on the imaging devices used, which is common in natural image datasets. Such sensitivity makes it difficult  to generalize to the specific natural image application without fine-tuning.
> > >
> > > - Semantic limitations of SA-1B pretraining: SAM is pre-trained on class-agnostic datasets (SA-1B) and produces segmentations without semantic understanding. However, many natural image applications *require semantic awareness* segmentation of multiple categories simultaneously, which cannot be directly achieved by prompt-based SAM without further adaptation.
> > >
> > > Recently, some works [2, 3] have attempted to fine-tune SAM to adapt to real-world applications  by composing large-scale, richly annotated datasets that separately target the two aspects discussed above, to improve its performance in the natural domain. In such scenarios, obtaining high-quality, pixel-level annotations remains costly and time-consuming. This aligns well with the strategy-centric goal of semi-supervised learning (SSL): to effectively utilize abundant unlabeled data (scaling the dataset to a larger scale) to enhance model adaptation and improve segmentation quality.
> > >
> > > We evaluate AgScore on standard benchmarks primarily to ensure fair comparison with  a wide range of recent well-established SSL methods. We sincerely appreciate your comment, and we will actively pursue this direction by establishing SSL benchmarks tailored for foundation model settings, which we believe will further expand the impact and applicability of the  SSL research community.
> > >
> > > Once again, thank you for your support of our work! If you have further questions, we would be happy to continue the conversation — after all, exchanging ideas is always *valuable*, regardless of the domain :)
> > >
> > > [1] Robustness Analysis on Foundational Segmentation Models.
> > >
> > > [2] Segment Anything in High Quality.
> > >
> > > [3] Semantic-SAM: Segment and Recognize Anything at Any Granularity.

---

### Official Review · Reviewer_bo7S · 2025-03-13

**Overall Recommendation:** 4

**Summary:**

This paper focuses on the confidence-based scoring functions in the semi-supervised semantic segmentation task. An agent construction strategy, aka., AgScore, is proposed to build clean sets of correct and incorrect pseudo labels. Experiments on three datasets show performance improvements in semi-supervised semantic segmentation.

**Claims And Evidence:**

The motivations from the "homogeneous pattern" is clearly shown in Fig.1.

Why the concept o Agent introduced here? It is not clear what is the relationships between "Agent" and the pseudo labels. The reviewer believes the name is somehow misleading.

**Essential References Not Discussed:**

N/A

**Experimental Designs Or Analyses:**

The experimental results mainly focus on the performance, while neglecting the training and evaluation efficiency, there should be a table or a subsection to discuss about this.

**Methods And Evaluation Criteria:**

The manuscript includes a substantial amount of theoretical analysis that appears to be tangential to the core focus of the study. Specifically, the exact operational mechanism of the 'agent score' metric remain unclear. The reviewer finds it challenging to identify how this metric is computed or applied in practice.

**Other Comments Or Suggestions:**

There can be some visualization results to prove the effectiveness of the proposed method, such as TSNE, etc.

**Other Strengths And Weaknesses:**

- **Formula Derivation**:
  - By analyzing the distribution of \( Z = \frac{Y}{X} \), it is proven that increasing the number of negative agent samples \( M \) can improve the separability between correct and incorrect pseudo-labels.
  - The relationship between FPR (False Positive Rate) and \( M \) is derived, proving that \(\frac{\partial FPR}{\partial M} < 0\).

- **Issues**:
  - The derivation assumes \( p_1 > p_2 \) and \( q_1 < q_2 \), but no experimental or theoretical support is provided to justify these assumptions.
  - The derivation does not consider that increasing \( M \) may lead to semantic overlap among negative agent samples, thereby affecting separability. This point is mentioned in the experimental section but is not discussed in detail in the theoretical analysis.

**Questions For Authors:**

1. **Training and Evaluation Efficiency**:
   - The experimental results primarily focus on performance metrics. Could the authors provide a table or subsection discussing the training and evaluation efficiency of the proposed method? This would help readers understand the computational cost and scalability of the approach.

2. **Assumptions in Theoretical Analysis**:
   - The derivation assumes \( p_1 > p_2 \) and \( q_1 < q_2 \). Could the authors provide experimental or theoretical evidence to support these assumptions? This would strengthen the validity of the theoretical analysis.

3. **Semantic Overlap in Negative Agents**:
   - The derivation does not consider the potential semantic overlap among negative agent samples when \( M \) is increased. Could the authors elaborate on how this might affect the separability and overall performance of the method?

4. **Visualization of Results**:
   - Could the authors include visualization results, such as t-SNE plots, to demonstrate the effectiveness of the proposed method? Visual evidence could provide additional insights into how the method distinguishes between correct and incorrect pseudo-labels.

5. **Impact of Negative Agent Selection**:
   - How does the selection strategy for negative agents (e.g., orthogonal selection) impact the overall performance? Could the authors provide a more detailed analysis or comparison with other selection strategies?

**Relation To Broader Scientific Literature:**

A new perspective on the pseudo label selection in semi-supervised semantic segmentation.

**Theoretical Claims:**

1. In the theoretical analysis, please discuss the independence assumption of X and Y and analyze its possible impact.

2. Suggestions: the theoretical analysis should supplement the discussion of the problem that the increase of negative sample agents may lead to semantic overlap, and analyzes its impact on separability.

---

> ### Author Rebuttal · Authors · 2025-04-01
>
> Thanks for taking the time to share your comments in the review assessment, as well as for acknowledging the **new perspective**,  **well-supported motivation** and  **clear idea**. We provide a detailed point-by-point response to your comments.
>
> Note that the following **link** refers to
> https://anonymous.4open.science/r/AgScore/Rebuttal.pdf
>
> ---
>
> **Q1**: Concept of Agent.
>
> **A1**: Sorry for not providing you with an intuitive understanding of the term "agent". In our work, "agents" refer to the sets of pixels that are considered to have correct or incorrect pseudo-labels, serving as a **bridge** for evaluating the correctness of the pseudo-labels for the remaining pixels. This is why "agents" are named.
>
> ---
>
> **Q2**: Independence Assumption of X and Y.
>
> **A2**: We assume that X and Y are independent for the convenience of subsequent theoretical derivations. In practice, we first select the top-1% and bottom-1% confidence pixels and then randomly/orthogonally sample positive/negative agents from them. This is done to **ensure that the independence assumption** holds to the greatest extent. To make the theoretical derivation more general, we even consider the case where the sampled agents are not identically distributed, i.e., the Poisson binomial distribution (Sec. C in the Appendix). Overall, our assumption is mild and supported by implementation.
>
> ---
>
> **Q3**: Value of our Theoretical Analysis.
>
> **A3**: Our theoretical analysis is closely intertwined with our method and experiments. As mentioned above, the agent selection strategy employed in our method is designed to ensure the independence assumption in the theoretical analysis. Moreover, the conclusion drawn from our theoretical analysis, "increasing M enhances the separability between correct and incorrect pseudo-labeled pixels," is also validated by Tab. 5 in our experiments. At the same time, the experiments also demonstrate that when M becomes excessively large, the performance degrade, since the independence assumption breaks down.
>
> ---
>
> **Q4**: Training and Evaluation Efficiency.
>
> **A4**: Thanks for your valuable suggestion. Notably, our AgScore is only involved in the selection of pseudo-labels during training and **does not introduce any additional cost during evaluation**. To further quantify the efficiency impact of AgScore, we report the GPU memory and training time of AgScore and baseline UniMatch under the same setting (Pascal VOC classic, 92 partition, cropsize=513, batchsize=8, ResNet50).
>
> |          | mIoU | GPU Memory (G) | Training Time (h) |
> | :------: | :--: | :------------: | :---------------: |
> | UniMatch | 67.4 |      44.2      |       23.6        |
> | AgScore  | 69.4 |      46.9      |       25.3        |
>
> We observe that AgScore brings a significant performance improvement at the cost of a slightly increased memory consumption and an acceptable increase in training time.
>
> ---
>
> **Q5**: Assumptions in Theoretical Analysis.
>
> **A5**: As shown in Fig. 1(c), we compute the average similarity between pixels corresponding to correct and incorrect pseudo-labels. It is evident that the similarity between pixels with correct pseudo-labels is greater than that between pixels with correct and incorrect pseudo-labels, i.e., $p_1 > p_2$ holds statistically. Similarly, $q_1 < q_2$ also holds statistically.
>
> ---
>
> **Q6**: Semantic Overlap in Negative Agents.
>
> **A6**: As you mentioned, when M becomes sufficiently large, there will be semantic overlap among negative agents. In this case, the most harmful negative agents are those that have semantic overlap with positive agents, as this violates the assumption that $p_1 > p_2$. Consequently, $\mu_1 - \mu_2$ increases, leading to a larger $\text{FPR}_\lambda$ (as shown in Lemma 4.3), which in turn degrades the separability between pixels with correct and incorrect pseudo-labels.
>
> ---
>
> **Q7**: More Visualization of Results.
>
> **A7**: As shown in **link** Fig. 4 and 5 , we supplement additional t-SNE visualizations for more classes on the Pascal and Cityscapes datasets. It can be observed that all classes clearly exhibit the Homogeneous Pattern. Additionally, as shown in **link** Fig. 7, the precision of the positive and negative agents obtained using simple top/bottom confidence is satisfactory. Therefore, leveraging relatively clean positive and negative agents, we are able to distinguish between correct and incorrect pseudo-labels.
>
> ---
>
> **Q8**: Impact of Negative Agent Selection.
>
> **A8**: In Tab. 6 of Appendix, we explore various strategies for negative agent selection strategy, including "Uniform" represents uniformly sampling negative agents from the candidate set, "Bottom" represents sampling negative agents in ascending order of confidence. We observe that our strategy of "Orthogonal" achieves the best results with a light computational cost.
>
> ---
>
> We hope our response can resolve your concern. Please do not hesitate to let us know if you have further questions :)

---

> > ### Comment · Reviewer_bo7S · 2025-04-01
> >
> > Thank the author for the thoughtful and detailed rebuttal. The reviewer sincerely appreciate the time and effort authors put into addressing the concerns. After carefully reviewing your responses, The reviewer is pleased to see that all of doubts have been resolved.

---

> > > ### Author Response · Authors · 2025-04-03
> > >
> > > Dear Reviewer bo7S,
> > >
> > > We sincerely appreciate your endorsement of our work and your positive feedback! We will make every effort to further improve our work!
> > >
> > > Authors

---

### Official Review · Reviewer_zrAv · 2025-03-14

**Overall Recommendation:** 3

**Summary:**

This study focuses on semi-supervised semantic segmentation which struggles to effectively use unlabeled data due to challenges in balancing true and false positives when filtering pseudo labels. This paper introduces an agent construction strategy and the Agent Score function (AgScore) to better identify correct and incorrect pseudo labels by leveraging patterns in the embedding space. AgScore is theoretically analyzed and shown to enhance segmentation performance across multiple frameworks and datasets, with code and models provided for future research.

**Claims And Evidence:**

No. The following claims made in the submission are not supported by clear and convincing evidence,

- Claim1: "pixels belonging to similar patterns tend to share homogeneous semantics compared to different patterns.“  Reason: The author uses the example in Figure 1(b) to demonstrate the homogeneous pattern, but this example is not representative and is likely the result of careful selection. More categories and a more comprehensive demonstration are needed to substantiate this claim. Moreover, although the features are distinctive, it is unconvincing that they are all classified as bicycles. The author needs to visualize the features of all relevant categories in the dataset together to enable reviewers to better assess whether Figure 1(b) is reasonable.

- Claim2: "However, this scoring function tends not to be preferred ascribed to the inherent trade-off between the true positive rate (TPR) and false positive rate (FPR), as illustrated in Figure 1 (a)." Reason: It is unclear how Figure 1(a) was generated.

- Claim3: "The results indicate that for any given pixel, there exists a higher probability of being correctly predicted if it exhibits a higher similarity to the set of correct pseudo labels compared to the set of incorrect pseudo labels." Reason: The same as Claim1, the evidence from Figure 1 is not convincing.

**Essential References Not Discussed:**

Perhaps, adding some more recent works in the related work section and including a dedicated section to discuss approaches for better pseudo-label planning would be beneficial.

**Experimental Designs Or Analyses:**

Refer to "Methods And Evaluation Criteria".

**Methods And Evaluation Criteria:**

This paper introduces an agent construction strategy and the Agent Score function (AgScore) to better identify correct and incorrect pseudo labels by leveraging patterns in the embedding space. However, the experiments lack sufficiently comprehensive ablation studies to evaluate the proposed method. For example,

- Why "we randomly select N pixels from the top-1% % confidence pixels in F in a class-balanced manner"? Any experiments to show the effectiveness of this method? How would the method handle a long-tailed distribution dataset? Additionally, is it possible to leverage the known ground truth (GT) data to construct this? I believe this part lacks corresponding experiments.

- Also, the design of section 3.3 lacks corresponding experiments.

- Is the designed orthogonal selection strategy reasonable, and would different training methods affect this approach (e.g., using different loss functions or different pixel category determination algorithms)?

- Whether the pixel selection strategy in the "Agent Construction" section is truly reasonable needs to be validated through more quantitative and qualitative experiments. For example, how do the distributions of predictions and ground truth (GT) in the top-1% and bottom-1% compare, and is there a difference compared to the top-2%? Is it necessary to rely on different datasets to rigorously control this parameter, and so on?

**Other Comments Or Suggestions:**

There are basically no typos in the paper that affect reading comprehension.

**Other Strengths And Weaknesses:**

Algorithm 1 and Algorithm 2 can be fully merged.

**Questions For Authors:**

Refer to "Methods And Evaluation Criteria" and "Claims And Evidence".

**Relation To Broader Scientific Literature:**

It is uncertain, as it is unclear whether the algorithm has limitations, such as being applicable only to specific datasets or particular domains.

**Theoretical Claims:**

I have reviewed it, and while there are no major issues, it seems that these validations do not address the core problems present in the paper.

---

> ### Author Rebuttal · Authors · 2025-04-01
>
> Thanks for taking the time to share your comments in the review assessment. We provide a detailed point-by-point response to your comments.
>
> **link** refers to https://anonymous.4open.science/r/AgScore/Rebuttal.pdf
>
> ---
>
> **Q1:** More Evidence.
>
> **A1**:
>
> - Claim 1: To demonstrate the generalization of the homogeneous pattern phenomenon, we conduct experiments and visualizations on both Pascal VOC and Cityscapes (1/16 partition) using ResNet-50.
>
>     (1) Single-category visualizations (Fig. 5 & 6 in link) highlight the top-4 most frequently predicted classes in each dataset. These results show that pixels with similar patterns within a class tend to share homogeneous semantics, while those from different classes exhibit distinct patterns. This supports our claim that “pixels with similar patterns tend to share homogeneous semantics compared to different patterns.”
>
>     (2) Multi-category visualizations offer a global perspective: correctly predicted pixels (darker) and incorrectly predicted ones (lighter) form well-separated clusters in the embedding space, further validating the link between visual patterns and semantic consistency.
>
> - Claim 2: To clarify how Fig. 1(a) is generated, we provide a step-by-step explanation in Fig. 7 (link).
>
>     Step (1) shows the confidence distributions of correct and incorrect predictions on Pascal VOC (1/16 split).
>
>     Step (2) defines true/false positive rates (TPR/FPR), while Steps (3) and (4) illustrate the trade-off between them under different confidence thresholds. Ground truth for unlabeled data is used here only for analysis.
> - Claim 3: In Fig. 1(c) (or Tab. 7 in link), we quantitatively measure the embedding similarity between pixels with correct and incorrect pseudo labels. For each class, we compute the similarity between each predicted pixel and the sets of correct/incorrect pixels, average the results, and normalize them class-wise.
>
> The results show that correctly predicted pixels have much higher average similarity to other correct pixels (0.925) than to incorrect ones (0.075), and vice versa. These findings provide numerical evidence that aligns with the qualitative observations in Fig. 1(b).
>
> ---
>
> **Q2**: More Experiments.
>
> **A2:**
> - Exp 1: Agent Selection Effectiveness & Class Imbalance
>
>     (1) We evaluate the top-1% confidence-based selection strategy on Pascal VOC (1/16 split, ResNet-50). As shown in Fig. 8 (link), the high precision of selected pixels for both positive and negative agents confirms the effectiveness of our confidence-based strategy.
>
>     (2) Pascal VOC itself exhibits severe class imbalance (e.g., background has >200× pixels than bicycle). Cityscapes and COCO show even larger head-to-tail ratios (>400 and >10,000). Despite this, our agent selection strategy (Algorithm 1), which samples clean examples in a class-balanced manner, remains robust—consistently improving performance across datasets (Tables 1–4).
>
>     (3) As for using labeled GT to construct agents, we’ve tested this on Pascal VOC (92 labeled images) with UniMatch as baseline (Tab. 10 in link). Due to the known distribution gap between labeled and unlabeled data (as noted in DAW, SoftMatch), agents built from GT-labeled pixels underperform compared to our confidence-based method.
>
> - Exp 2: Design Choices for AgScore
>
>    (1) Incorporating Negative Agents: Using only positive agents (Row 1) omits key contrastive signals, leading to inferior performance. Adding negative agents (Rows 2/3) in a proportional form introduces nonlinearity that normalizes and stabilizes similarity scores, enhancing label separability.
>
>    (2) Exponentiation for Score Scaling: Applying the exponential function to cosine similarity (Row 3) amplifies score differences, further improving performance—aligning with our theoretical motivation.
>
> - Exp 3: Orthogonal Selection Strategy
>
>     (1) Our orthogonal selection strategy is tailored for negative agent construction, selecting diverse, low-confidence samples to ensure agent purity (Fig. 8).
>
>    (2) This enhances semantic coverage and improves the separation between correct/incorrect predictions, supporting our theoretical design.
>
>     (3) In practice, AgScore serves as a general plug-in compatible with various methods (e.g., with different loss functions), making our orthogonal selection both reasonable and flexible.
>
> - Exp 4: Sensitivity to Selection Ratio Experiments in Fig. 9 (link) show that raising the pixel selection ratio to 2% reduces accuracy. Our 1% threshold is thus a simple yet effective default. While not heavily tuned, this setting offers room for further optimization based on dataset characteristics.
>
> We will include this analysis in the revised version.
>
> ---
>
> **Q3:** Theory.
>
> **A3:** Please refer to the answer **A3** of reviewer bo7S.
>
> ---
>
> **Q4:** Others.
>
> **A4:** We will improve the typos and add more discussions due to space limitations.
>
> ---
>
> We hope our response can resolve your concern. Have a nice day!

---

### Official Review · Reviewer_ZpSC · 2025-03-28

**Overall Recommendation:** 3

**Summary:**

The authors introduce “AgScore” (Agent Score), a scoring function to filter out unreliable pseudo labels at the pixel level in order to improve the performance of existing semi-supervised semantic segmentation (SSSS) methods. Unlike prior work that primarily relies on high-confidence thresholding in the prediction (logits) space to decide which pseudo labels to use, this paper focuses on the phenomenon of “homogeneous pattern” in the feature/embedding space. Specifically, the authors first construct two sets of agents: positive agents (high-confidence, presumably correct pseudo labels) and negative agents (low-confidence, presumably incorrect pseudo labels). chosen through an orthogonal selection strategy ensuring semantic diversity. Then, a function computes each unlabeled pixel’s similarity to these two agent sets. If a pixel is more similar to the positive agent set than the negative agent set, the method deems its pseudo label more reliable; otherwise, it is filtered out. Empirically, integrating AgScore into existing SSSS frameworks such as FixMatch, UniMatch, and RankMatch consistently improves segmentation performance across popular benchmarks like Pascal VOC, Cityscapes, and COCO.

**Claims And Evidence:**

The core claim that incorporating feature-space similarity (via positive and negative agents) enables more precise filtering of pseudo labels than standard confidence-thresholding alone is well supported by the experiments conducted integrating AgScore into three well-known baselines.

**Essential References Not Discussed:**

Overall, references to major prior SSSS methods (e.g., FixMatch, MeanTeacher, ST++, ReCo, etc.) are included. Even very recent ones like RankMatch (2024).

**Experimental Designs Or Analyses:**

These are sound as they follow the general practices in the field.

**Methods And Evaluation Criteria:**

* Evaluation was done on three standard SS datasets (Pascal VOC, Cityscapes, COCO) using various data partition protocols following the practices in the field.
* AgScore is integrated into three different SS frameworks with improved performances in all cases, proving its effectiveness.
* Ablation studies (Tables 5, 6 and Figure 3 from the supplementary) support the design choices.

**Other Comments Or Suggestions:**

I recommend the authors have a closer look at their paper, there are still some typos to deal with and paragraphs ("trad-off") that need rephrasing (Page 1, column 2, L45-48) to improve clarity (Table 1 mentions bolded results, but none of them are bolded). But the most alarming thing is the confusion the others make between AgScore and AugSeg (a method published and referenced in CVPR 2023), which the authors mentioned as their own when here the authors should have referred to AgScore. I treated this as an error; otherwise, using "Our AugSeg" breaches the double-blind submission restriction.

**Other Strengths And Weaknesses:**

• Strengths:
  - The agent construction idea is simple but effective and can be integrated with minimal overhead into multiple SSSS frameworks.
  - The experimental validation systematically shows improvements.
  - The theoretical discussion is a welcome addition.

• Weaknesses:
  - The approach is based on carefully sampling from the top/bottom confidence. Though the paper includes ablations, there might be edge cases where strong multi-class overlap or severely noisy unlabeled sets pose challenges.
  - The approach’s success depends on the assumption that extremely high-confidence pixels are reliably correct. This is usually true for well-trained teacher networks but might occasionally fail early in training or with domain-shift data.

**Questions For Authors:**

What it's currently missing from the experiments is a proper assessment of AgScore's robustness when the teacher model is underfitted in the early stage of training. Are there any warm-up strategies to ensure reliable positive agents from the start? Also, I am curious to know this, especially for domain shifts in a semi-supervised setting. Does the "top confidence = correct" assumption still hold there? Any insights from the authors are welcomed.

**Relation To Broader Scientific Literature:**

* The authors place their work in the context of existing semi-supervised segmentation approaches that rely on teacher-student frameworks, including FixMatch and others. They also connect their approach to contrastive learning in the feature space or methods that propose robust pseudo-label selection.
* The authors also relate their approach to relevant classical SSL ideas (like consistency regularization and pseudo-label thresholding).

**Theoretical Claims:**

I have not properly assessed the correctness of the theoretical claims and the proofs, but I confirm I have looked over them, and nothing evidently stands out as wrong.

---

> ### Author Rebuttal · Authors · 2025-04-01
>
> Thanks for taking the time to share your comments in the review assessment, as well as for acknowledging the **well-supported core claim**, **simple but effective idea**, **systematic experimental validation**, and **insightful theoretical analysis**. We provide a detailed point-by-point response to your comments.
>
> Note that the following **link** refers to
> https://anonymous.4open.science/r/AgScore/Rebuttal.pdf
>
> ---
>
> **Q1**: Robustness of AgScore in handling edge cases.
>
> **A1**: We appreciate your insightful question. Indeed, edge cases like **large domain shift**, **strong multi-class overlap** or **severely noisy unlabeled data** pose significant challenges to semi-supervised learning. To mitigate these limitations, our orthogonal selection strategy for constructing the negative agent set aims to select representative and diverse negative samples, covering a broader semantic space of incorrect pseudo-labels. This helps to some extent in handling domain shift, class overlap and noisy data. However, we acknowledge that in extremely severe edge cases, the baseline methods may struggle to produce effective predictions, which could hinder the selection of reliable agents and the evaluation of pseudo-labels.
>
> In this paper, we only focus on the conventional semi-supervised setting, where the baseline methods generally yield effective predictions that support our approach. Your valuable suggestion inspires us to explore extending our method to extreme cases in future work, which is crucial for enhancing the robustness of semi-supervised segmentation.
>
> ---
>
> **Q2**: Robustness of AgScore when the teacher model is underfitted.
>
> **A2**: Indeed, the key to the success of our method lies in selecting reliable positive and negative agents. As shown in Fig. 9 in **link**, we record the precision of the selected positive and negative agents after each training epoch. The agents are selected based on the top-1% and bottom-1% confidence scores, respectively. Notably, even after just the 0-th epoch, the precision of both positive and negative agents is already satisfactorily high. This suggests that the teacher model can provide reliable agents for AgScore almost right from the start of training. Consequently, to maintain the simplicity of AgScore, we do not employ any specific warm-up strategies, as the experiments demonstrate its effectiveness without such strategies. We concur that carefully integrating warm-up strategies could potentially lead to better performance, and we leave this as a direction for future investigation.
>
> ---
>
> **Q3**: Typo and "AugSeg" misuse.
>
> **A3**: Thanks for your meticulous review and valuable feedback. We apologize for the oversight in mistakenly referring to "AgScore" as "AugSeg" in the Experiments section. We will carefully review the manuscript, correct these errors, and improve the clarity of our expressions.
>
> ---
>
> We hope our response can resolve your concern. Please do not hesitate to let us know if you have further questions :)

---

### Decision · Program_Chairs · 2025-05-01

**Decision:**

Accept (poster)

**Comment:**

This paper received mixed reviews initially. After rebuttal, three reviewers increase their scores, and now all reviewers lean to accept the paper. Two reviewers still have some concerns. One has a concern about the value of semi-supervised semantic segmentation in the era of large models, e.g., SAM; The other has a concern about the reliability of the proposed method. The AC recommends acceptance, but the authors are required to address these concerns in the camera-ready version.